# Cognitive regulation alters social and dietary choice by changing attribute representations in domain-general and domain-specific brain circuits

Anita Tusche[1]*, Cendri A Hutcherson[2,3]

[1]Division of the Humanities and Social Sciences, California Institute of Technology, Pasadena, United States; [2]Department of Psychology, University of Toronto Scarborough, Toronto, Canada; [3]Department of Marketing, Rotman School of Management, University of Toronto, Toronto, Canada

**Abstract** Are some people generally more successful using cognitive regulation or does it depend on the choice domain? Why? We combined behavioral computational modeling and multivariate decoding of fMRI responses to identify neural loci of regulation-related shifts in value representations across goals and domains (dietary or altruistic choice). Surprisingly, regulatory goals did not alter integrative value representations in the ventromedial prefrontal cortex, which represented all choice-relevant attributes across goals and domains. Instead, the dorsolateral prefrontal cortex (DLPFC) flexibly encoded goal-consistent values and predicted regulatory success for the majority of choice-relevant attributes, using attribute-specific neural codes. We also identified domain-specific exceptions: goal-dependent encoding of prosocial attributes localized to precuneus and temporo-parietal junction (not DLPFC). Our results suggest that cognitive regulation operated by changing specific attribute representations (not integrated values). Evidence of domain-general and domain-specific neural loci reveals important divisions of labor, explaining when and why regulatory success generalizes (or doesn't) across contexts and domains.
DOI: https://doi.org/10.7554/eLife.31185.001

*For correspondence:
anita.tusche@gmail.com

**Competing interests:** The authors declare that no competing interests exist.

## Introduction

Choices often require us to weigh competing considerations. Does a decadent piece of cake merit the pounds we'll put on afterwards? Should the pleas of a homeless person trump our own selfish needs? Empirical evidence suggests that the answer to these questions depends in part on a decision maker's goals (*Bettman et al., 1998*) and can be affected by intentional control (*Hare et al., 2011a*; *Hutcherson et al., 2012*; *Sokol-Hessner et al., 2013*). Cognitive regulation of decision making thus serves an important function in goal-directed behavior (*Magar et al., 2008*), relying on attention, working memory, and executive control to promote particular, goal-congruent choices (e. g., eat healthier, be kinder). Cognitive regulation of decision making is an important technique in therapeutic interventions for problematic behaviors, including obesity (*Shaw et al., 2005*), addiction (*Carroll and Onken, 2005*), and other decision making disorders (*Sylvain et al., 1997*). Previous findings have significantly advanced our understanding of the psychological and neural bases of cognitive regulation of decision making (*Hare et al., 2011a*; *Hutcherson et al., 2012*; *Hare et al., 2009*; *Kober et al., 2010*), yet important questions about its computational underpinnings remain. At what level of the processing stream does goal-dependent cognitive regulation change the typical trajectory of choice? Does it operate in the same manner in different contexts, or does it depend on the domain? Answering these questions has important ramifications for understanding when people

succeed or fail to implement their regulatory goals during decision making, why some people seem to succeed more often than others, and whether there are neural targets for treatment or biomarkers to identify at-risk individuals.

In studies of basic choice, weighted additive utility models have been used successfully to capture patterns in human behavior across a variety of domains (*Charness and Rabin, 2002*; *Schoemaker and Waid, 1982*). In these models, decision makers compute the decision value (DV) of each option as the weighted sum of its choice-relevant attributes $\left(DV = \sum_i w_i Attribute_i\right)$ (*Keeney and Raiffa, 1993*; *Anderson et al., 1971*) and compare them to make a choice. Recent neuroscience work provides evidence in favor of this model, observing signals related to the value of specific attributes in distinct cortical and subcortical areas, for both social (*Hutcherson et al., 2015a*; *Hutcherson et al., 2015b*) and non-social choices (*Lim et al., 2011*; *Basten et al., 2010*). In turn, signals correlated with the overall, integrated decision value of an option have been observed in multiple regions, such as the ventromedial prefrontal cortex (VMPFC) and ventral striatum (*Hutcherson et al., 2012*; *Plassmann et al., 2007*; *Kable and Glimcher, 2007*; *Knutson et al., 2007*; *Grueschow et al., 2015*; *Bartra et al., 2013*; *Clithero and Rangel, 2014*). A key goal of neuroeconomics is to describe how these attribute and decision value computations change as a function of regulatory goals and contexts, and to link such changes to regulatory success. Here, we sought to address three important questions about this process.

First, at what level does cognitive regulation operate to change value representations? Based on the neuroeconomic model outlined above, we hypothesized two possibilities. The *attribute-level* hypothesis suggests that cognitive regulation of decision making could alter value representations at a relatively low level, by amplifying or diminishing attribute representations directly in a distributed set of specific, dedicated attribute-coding areas, similar to attentional effects on visual object encoding (*Egner and Hirsch, 2005*). Alternatively, the *integration-level* hypothesis suggests that cognitive regulation of decision making might operate at comparatively higher levels in centralized, domain-general value integration areas such as the VMPFC (*Hare et al., 2009*; *Hare et al., 2010*).

Second, we aimed to explicitly test whether cognitive regulation alters value representations at the *same* level regardless of domain, or whether it differs as a function of attributes, goals or choice domain. For example, some attributes (such as taste) may be innate and prepotent, while other attributes (such as health or social considerations) may be more abstract or effortful to construct (*Sullivan et al., 2015*; *Metcalfe and Mischel, 1999*; *Loewenstein and Small, 2007*). We sought to test whether these distinctions might affect where and how cognitive regulation operates to alter value representations during decision making. We also sought to determine whether this translates into distinct regulatory capacities as a function of regulatory goal or choice domain.

Finally, we sought to shed light on whether information represented in VMPFC and dorsolateral prefrontal cortex (DLPFC) supports either attribute-level or integration-level changes in value during cognitive regulation of decision making. For example, some experimental evidence supports the idea that the DLPFC might represent more abstract attributes like health (*Hare et al., 2011b2011*; *Bhanji and Beer, 2012*), and that regulatory control could modulate interactions between the DLPFC and VMPFC to change attribute weights in integrative decision value computations (*Hare et al., 2011a*; *Sokol-Hessner et al., 2013*; *Kober et al., 2010*; *Bhanji and Beer, 2012*). However, several failures to observe changes in the VMPFC during cognitive regulation of decision making (*Hutcherson et al., 2012*; *Hollmann et al., 2012*; *Yokum and Stice, 2013*) suggest the need to either measure value computation in a more sensitive way, or to identify alternate routes to behavioral change.

Addressing these issues requires investigating regulatory control across multiple attributes and domains, using a sophisticated array of approaches for identifying changes in the representations of both specific attributes and integrated value signals. We used functional magnetic resonance imaging (fMRI) to measure brain responses while subjects completed two choice tasks, separated in time by up to 24 months (*Figure 1A,B*). Choices involved foods varying in healthiness and tastiness (food task) or monetary proposals varying in payoffs for subjects and an anonymous partner (altruism task). To mimic the kinds of cognitive reframing approaches that are often used in therapy for decision making disorders (*Shaw et al., 2005*; *Carroll and Onken, 2005*), both tasks asked subjects to adopt distinct regulatory goals designed to highlight different choice attributes (e.g., 'focus on the food's healthiness', 'focus on your partner's feelings'). To pinpoint whether and how regulation altered specific attribute representations or integrative value computations at the behavioral and neural level,

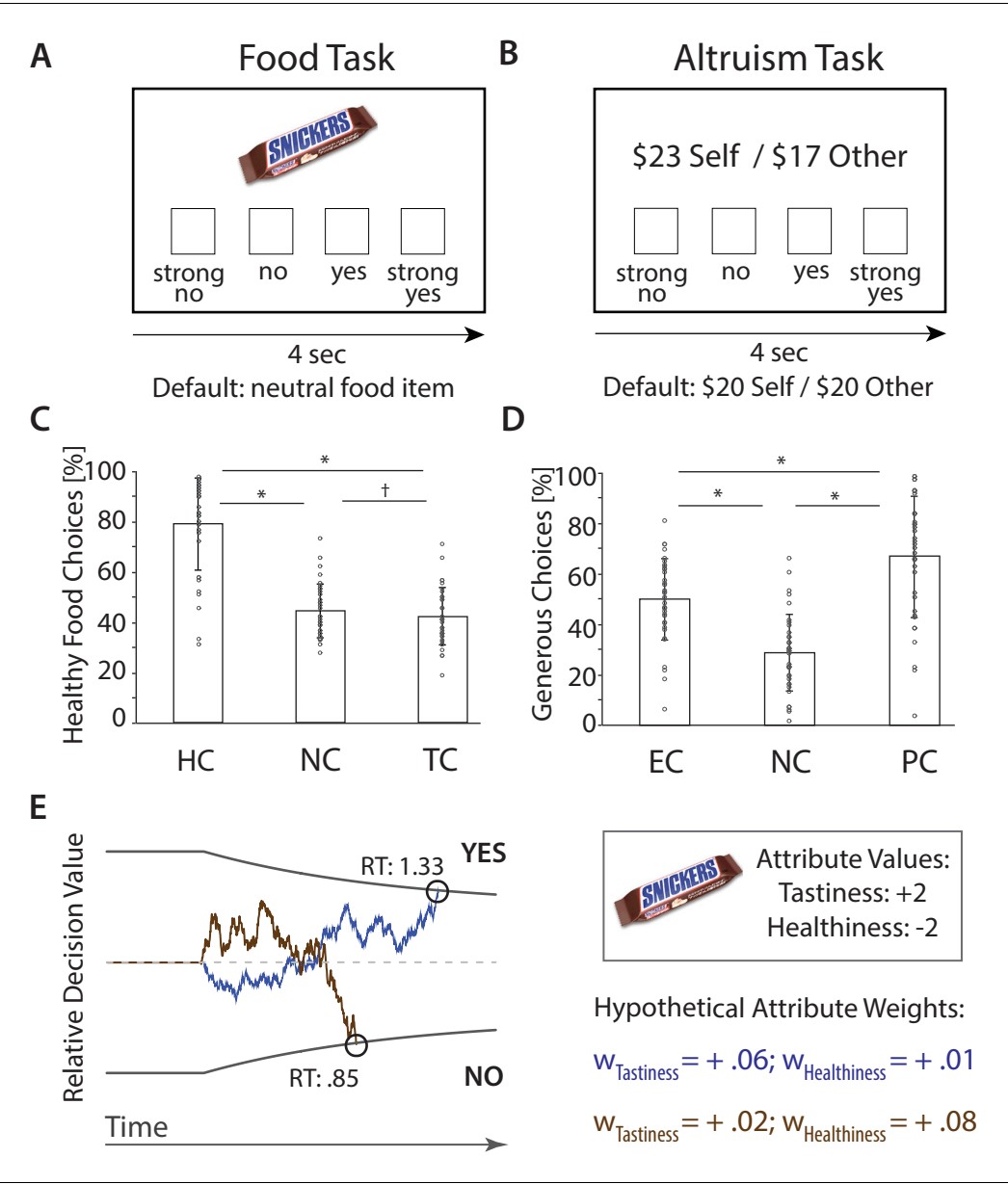

**Figure 1.** fMRI Paradigms and Choices. (**A**) Food Task. Subjects chose between on-screen food items that varied in tastiness and healthiness and a neutral default food. Choices were made in 'Natural' [NC], 'Focus on Health' [HC], and 'Focus on Taste' Conditions [TC]. (**B**) Altruism Task. Subjects chose between on-screen proposals that affected the payoff of themselves ($Self) and an anonymous partner ($Other) and a default option ($20 for both). Choices were made in 'Natural' [NC], 'Focus on Ethics' [EC], and 'Focus on Partner' Conditions [PC]. (**C**) (**D**). Bar plots illustrate condition-wise percentages of healthy (C) and generous (D) choices (M ± SD), and subject-specific scores (circles). *p < 0.05, corrected, †p < 0.05, uncorrected. (**E**) Computational behavioral model (DDM). Choices (yes/no) are made when the sequential accumulation of noisy value information that unfolds over time crosses the predefined upper or lower threshold for choice. The relative decision value (RDV) at a point in time (t) is computed as the weighted sum of choice relevant attributes plus noise ($\varepsilon$) (i.e., $RDV_t = RDV_{t-1} + w_{Tastiness}$ * Tastiness + $w_{Healthiness}$ * Healthiness + $\varepsilon_t$). In the example displayed here, the value of a candy bar will tend to accumulate in a positive direction if the weight on Tastiness is high (blue line), yielding a choice in favor of a tasty but unhealthy item. However, the value of the food item is more likely to accumulate in a negative direction if the weight on Healthiness is high (brown line). Note that saying Yes can sometimes indicate a healthy choice, and sometimes an unhealthy choice. (RT = reaction times [sec]; figure adapted from [**Hutcherson et al., 2015b**; **Adolphs and Tusche, 2017**]).

DOI: https://doi.org/10.7554/eLife.31185.002

*Figure 1 continued on next page*

*Figure 1 continued*

The following figure supplement is available for figure 1:

**Figure supplement 1.** Drift diffusion model (DDM) fits to behavior in both choice tasks.

DOI: https://doi.org/10.7554/eLife.31185.003

we combined a multi-attribute extension of the drift diffusion model (DDM) (*Ratcliff and McKoon, 2008*; *Smith and Ratcliff, 2004*) with multivariate pattern analyses (MVPA) of neural responses (*Kriegeskorte et al., 2006*; *Haynes and Rees, 2006*). MVPA approaches to fMRI data exploit information encoded across multiple voxels and have been suggested to detect information that would be missed by conventional univariate analyses (*Kriegeskorte and Bandettini, 2007*). Past research on cognitive regulation has relied primarily on mass univariate approaches, which could account for some of the inconsistencies observed in the literature. Our study used MVPA to examine whether and how directed attention to specific goals affects the neural information content (i.e., decoding accuracies) for attribute values in different social and non-social decision contexts. We hypothesized that goal-dependent changes in neural decoding accuracies would match predictions on altered attribute weights from the behavioral computational model. We investigated where such changes occurred, whether they operate in generic or domain-specific manner, and whether they predicted specific aspects of regulatory success across individuals.

## Results

### Behavior

To identify how value computations change to accommodate regulatory goals, our analysis strategy proceeded in the several steps. First, on the behavioral level, we confirmed that regulatory goals resulted in altered choice behavior. We also used our computational behavioral models (multi-attribute drift diffusion models, DDMs) to link these alterations to amplification or suppression of the influence of specific choice-relevant attributes on choices.

#### Choice behavior

Choices in both tasks varied considerably by regulatory goal (*Figure 1C,D*). In the food task, subjects made choices in three conditions: Respond Naturally [NC] ('respond as you naturally would'), Focus on Health [HC] ('focus on the healthiness of the food when making the choice'), and Focus on Taste [TC] ('focus on the tastiness of the food when making the choice'), implemented in interleaved blocks (see Appendix 1 – Instructions for regulatory conditions in both choice tasks for instructions). We defined a healthy choice as accepting the on-screen food if it was healthier than the default food (based on subject-specific healthiness ratings obtained outside of the scanner, see Materials and methods), and rejecting it otherwise. As expected, subjects made significantly healthier choices during HC (M ± SD: 78.83% ± 18.46) compared to both NC (44.31% ± 10.71) and TC (41.99% ± 11.46; paired t-tests: p's < 0.001, Bonferroni corrected unless stated otherwise). They also made marginally less healthy choices during TC than NC (p = 0.043, uncorrected; repeated measures ANOVA across all conditions: F(2,35) = 97.01, p < 0.001).

In the altruism task, subjects were instructed either to Respond Naturally [NC] ('respond as you naturally would'), Focus on Ethics [EC] ('focus on doing the right thing and consider the ethical or moral implications of your choice'), or Focus on Partner [PC] ('focus on your partner's feelings and how the other person is affected by your choice') (see Appendix 1 – Instructions for regulatory conditions in both choice tasks for instructions). We defined an altruistic choice as accepting an on-screen proposal whose outcome (relative to the default) benefitted the other at a cost to the self, or rejecting one in which the subject stood to benefit but their partner did not. As expected, subjects made altruistic choices significantly less often under NC (28.71% ± 15.48) compared to EC (49.94% ± 16.22) or PC (66.97% ± 24.35; p's < 0.001; F(2,35) = 65.96, p < 0.001) (*Figure 1D*). Altruistic choices were also significantly higher in PC than EC (p < 0.001), suggesting that directing attention to another persons' feelings generally increased altruism more effectively than considering social and moral norms. Overall, these findings confirmed that regulatory goals resulted in altered choice behavior in the food task and the altruism task.

**Table 1.** Correlation of regulatory success (RS) in both choice tasks.

| | | Regulatory Success (RS) in Food Task | | |
| --- | --- | --- | --- | --- |
| | | RS [HC - NC] | RS [HC - TC] | RS [NC - TC] |
| RS in Altruism Task | ΔRS [PC - NC] | 0.52 * | 0.56 * | 0.33 † |
| | ΔRS [EC - NC] | 0.37 † | 0.37 † | 0.14 |
| | ΔRS [PC - EC] | 0.48 * | 0.53 * | 0.38 † |

*p < 0.05 Bonferroni corrected, †p < 0.05 uncorrected; HC = Health Condition, NC = Natural Condition, TC = Taste Condition, PC = Partner Condition, EC = Ethics Condition

DOI: https://doi.org/10.7554/eLife.31185.004

## Regulatory success

Given the considerable individual heterogeneity in the extent of these changes, we also sought to understand whether this heterogeneity might be consistent across tasks and regulatory instructions. Regulatory success – defined as goal-consistent changes in percent healthy or altruistic choices (*Hare et al., 2011a*) (e.g., the increase in healthy choices during HC compared to NC) – covaried across tasks (*Table 1*). People who chose healthy foods more often when attending to a food's healthiness also behaved more altruistically when focusing on pro-social attributes. These results did not depend on the delay between tasks (partial correlations controlling for delay of up to 24 months, M ± SD: 16.42 ± 8.66, range: 1 to 24) or differences in baseline responding *within* a particular condition: the percentage of healthy and altruistic choices during NC blocks of both tasks did not correlate (all p's > 0.05, uncorrected). Instead, they were driven by choice behavior during regulation: healthy choice during HC correlated with altruistic choice in both EC (r = 0.45, p < 0.05) and PC (r = 0.66, p < 0.001). Overall, these findings indicate that an individuals' regulatory success generalized across choice domains. We found no significant correlation of self-reported motivation to comply with instructions with regulation success in the food task (all p's > 0.14, uncorrected) or the altruism task (all p's > 0.16, uncorrected) (Appendix 1 – Self-reported motivation to comply with instructions and observed regulation-success).

## Computational parameter estimates (DDMs)

We hypothesized that changes in choice behavior could result either from increased weighting of goal-consistent attributes (e.g. healthiness in HC), decreased weighting of goal-inconsistent attributes (e.g. tastiness in HC), or both. We tested these possibilities by fitting multi-attribute DDMs to behavior, separately for each subject in each condition and task (see Appendix 1 – Drift diffusion model for details). Model fits to behavior indicated that we were able to capture both choices and

**Table 2.** Model-estimated weights (w) assigned to choice-relevant attributes in the food task and altruism task (DDMs).

| Attributes | Sample size (N) | Regulation Conditions in Food Task | | |
| --- | --- | --- | --- | --- |
| | | Mean (±SD) | Mean (±SD) | Mean (±SD) |
| | | Natural [NC] | Focus on Health [HC] | Focus on Taste [TC] |
| w Healthiness | 36 | −0.0003 (±0.0040) | 0.0121 (±0.0074) | −0.0019 (±0.0037) |
| w Tastiness | 36 | 0.0163 (±0.0054) | 0.0044 (±0.0064) | 0.0167 (±0.0051) |
| | | Regulation Conditions in Altruism Task | | |
| | | Natural [NC] | Focus on Partner [PC] | Focus on Ethics [EC] |
| w $Self | 49 | 0.0082 (±0.0038) | 0.0037 (±0.0057) | 0.0070 (±0.0050) |
| w $Self | 36 | 0.0082 (±0.0040) | 0.0037 (±0.0059) | 0.0068 (±0.0049) |
| w $Other | 49 | 0.0010 (±0.0039) | 0.0059 (±0.0040) | 0.0047 (±0.0049) |
| w $Other | 36 | 0.0009 (±0.0039) | 0.0057 (±0.0040) | 0.0048 (±0.0049) |
| w Fairness | 49 | 0.0018 (±0.0034) | 0.0029 (±0.0035) | 0.0062 (±0.0056) |
| w Fairness | 36 | 0.0019 (±0.0035) | 0.0026 (±0.0036) | 0.0062 (±0.0058) |

DOI: https://doi.org/10.7554/eLife.31185.007

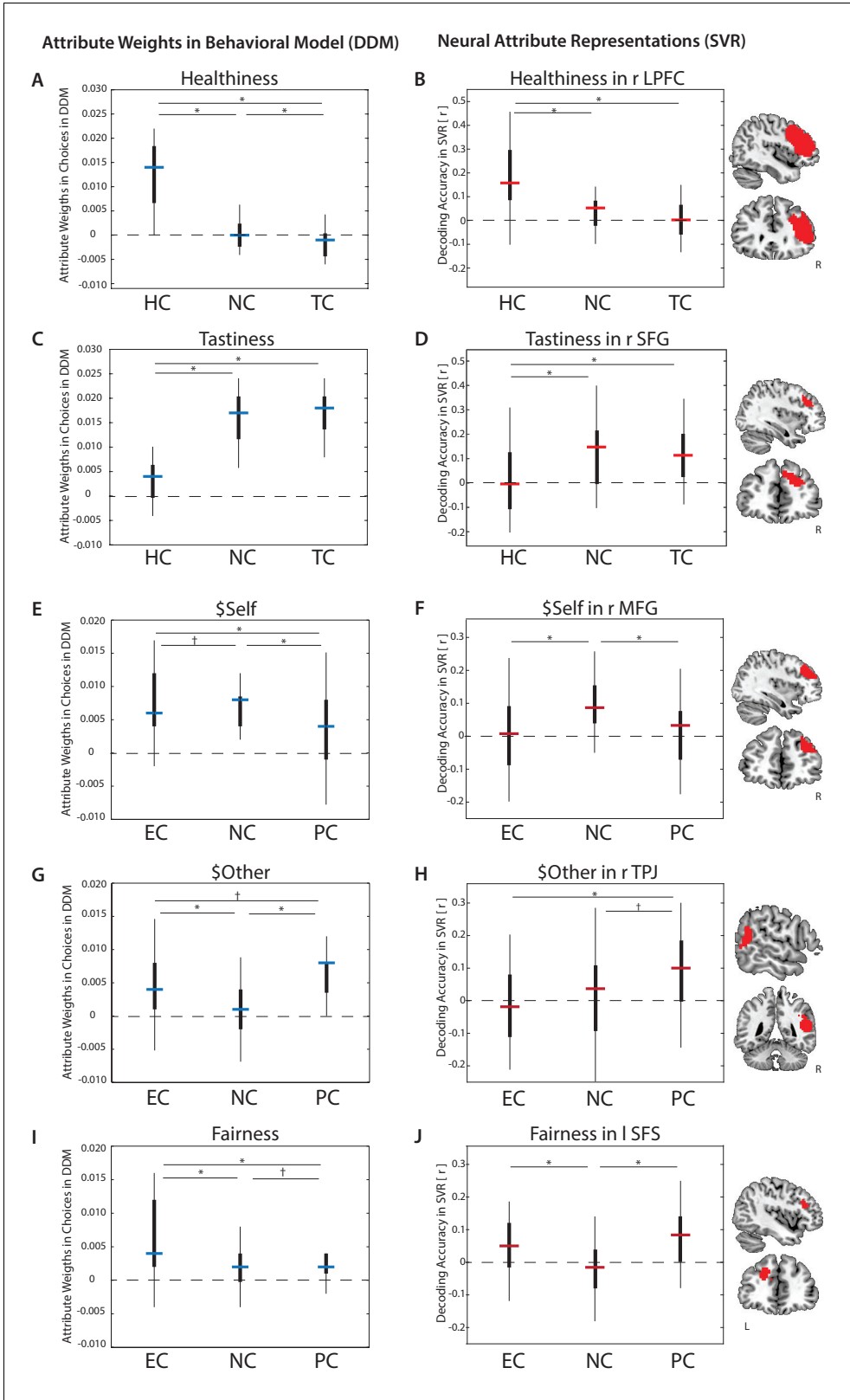

**Figure 2.** Goal-dependent modulation of attribute value encoding. *Behavioral* weights (left column) assigned to attributes in food choices (A. Healthiness, C. Tastiness) or altruistic choices (E. $Self, G. $Other, I. Fairness) varied by regulatory goal (estimates of drift diffusion models, DDMs). *Neural* decoding accuracies of attribute values (right column) also varied across conditions in specific brain regions (B. Healthiness, D. Tastiness, F. $Self, H. $Other, J. Fairness) ($p < 0.05$, FWE corrected at cluster-level) (estimates of Support Vector Regression models, SVRs). Bars represent median

*Figure 2 continued on next page*

*Figure 2 continued*

estimates (blue = behavioral DDMs, red = neural SVRs; black boxes signify 25–75 percentile, lines illustrate the overall distribution), HC = Health Condition, NC = Natural Condition, TC = Taste Condition, PC = Partner Condition, EC = Ethics Condition, L = left hemisphere, R = right hemisphere, LPFC = Lateral Prefrontal Cortex, SFG = Superior Frontal Gyrus, MFG = Mid Frontal Gyrus, TPJ = Temporoparietal Junction, SFS = Superior Frontal Gyrus.

DOI: https://doi.org/10.7554/eLife.31185.005

The following figure supplement is available for figure 2:

**Figure supplement 1.** Goal-dependent modulation of neural value encoding in DMPFC ($Self) and Precuneus ($Other) in the altruism task.

DOI: https://doi.org/10.7554/eLife.31185.006

RTs with high accuracy (*Figure 1—figure supplement 1*). Supplemental analyses also confirmed that the DDM did not perform worse in capturing behavior during regulation conditions compared to natural choices (Appendix 1 – Drift diffusion model). To determine if regulatory goals altered weights assigned to distinct attributes, we computed repeated measures ANOVAs with regulatory goal as a within-subject factor, separately for each attribute.

As predicted, regulatory goals in the food task changed the weights assigned to tastiness and healthiness (all $F(2,35) \geq 103.36$, p's < 0.001; see *Table 2* for attribute-specific estimates; for complete list of model-estimates and RTs see *Supplementary file 1A*). Healthiness influenced food choices *more* in HC, and *less* in TC, compared to NC (*Figure 2A*, all p's $\leq$ 0.001). By contrast, tastiness influenced food choices less in HC, compared to both NC (p < 0.001) and TC (p < 0.001) (*Figure 2C*). No differences emerged between NC and TC (p = 0.47, uncorrected, 2-tailed), suggesting that decision processes in TC likely resemble natural choice contexts.

Regulatory goals had a similarly dramatic influence on attribute weights in the altruism task (all $F(2,48) \geq 21.48$, p's < 0.001; *Table 2*). Subjects' choices were swayed more strongly by their own monetary outcome ($Self) in NC compared to PC (p < 0.001) and marginally compared to EC (p = 0.059, uncorrected, 2-tailed) (*Figure 2E*). Moreover, the influence of their own payoffs on choices decreased more dramatically in PC than EC (p < 0.001). In contrast, estimated weights on the partner's monetary outcome ($Other) increased for both pro-social regulatory conditions compared to NC (p's < 0.001), with marginally higher weights in PC than EC (p = 0.013, uncorrected, 2-tailed) (*Figure 2G*). Fairness of proposed payouts ($-1*$|$Self - $Other|) influenced choices significantly less in NC compared to EC (p < 0.001), and marginally less compared to PC (p = 0.021, uncorrected, 2-tailed). Weight on fairness was also significantly higher in EC than PC (p < 0.001) (*Figure 2I*). Note that within-task results for the altruism task are reported for the slightly larger sample size of 49 subjects. Considering only the subset of subjects that also participated in the food task (N = 36) yielded comparable weights for attributes in altruistic choices (*Table 2*). Overall, the results suggest that regulatory goals changed choice behavior by both increasing weighting of goal-consistent attributes (e.g. healthiness in HC) and decreasing weighting of goal-inconsistent attributes (e.g. tastiness in HC).

## Neural encoding of choice attributes and effects of regulation

Next, we examined neural underpinnings of goal-consistent increases/decreases in the influence of attributes on altered choices in both tasks. This analysis step was designed to provide evidence for the effects of regulation at the attribute-level or integration-level. Both hypotheses suggest that changes in the influence of distinct attributes on choice should correspond to changes in neural encoding of those attributes. However, they make different predictions about *where* these changes should be observed. The attribute-level hypothesis predicts that attributes are encoded in attribute-specific brain areas and that regulation should result in changes to these local representations. By contrast, the integration-level hypothesis suggests that attribute-specific areas should encode attributes similarly *regardless* of the regulatory goal. Instead, altered representations should appear only within centralized brain regions associated with value-integration, such as the VMPFC, and should be detectable in a common signal associated with integrated values. We tested these distinct predictions by examining where attribute values were represented in the brain, and how these representations varied as a function of regulatory focus. We also explicitly tested whether the locus of effect differed across attributes (e.g. tastiness/healthiness, $Self/$Other/Fairness) or choice domain (e.g. social, non-social).

## Neural encoding of choice attributes and decision values across conditions

Our behavioral results suggest that a weighted combination of different choice-relevant attributes captures behavior in both choice tasks (*Figure 1—figure supplement 1*), implying that attribute information should be represented in the brain. However, the generality and specificity of this encoding has important implications both for theories about how different attributes are constructed, and how regulation operates to modulate their influence. We first sought to determine which brain regions reliably encoded trial-by-trial variation in a given attribute across experimental conditions and goals. Thus, this first set of decoding analyses tested *if* neural activation patterns encode attribute values, irrespective of whether one or several conditions drive this predictive information. To this end, we averaged the condition-specific decoding maps of an attribute for each subject and tested for brain regions that reliably predict values of the attribute at the group level. Consistent with predictions, information about each attribute could be decoded significantly above chance in multiple brain regions (*Table 3*), including the VMPFC, and, for some attributes, the DLPFC. This was also true for trial-by trial encoding of decision values (DVs, corresponding to observable choices in the altruism and food task). See *Supplementary file 1B* (main effects) for a complete list of results and details on the clusters in the (V)MPFC and DLPFC for the neural decoding of DVs.

## Conjunction of neural representations of choice attributes

Given the robust coding of individual attributes, we asked whether any brain regions encoded *all* attribute values across all contexts, as might be expected of domain-general areas contributing to value integration processes. A formal conjunction of all attribute-specific decoding maps (Healthiness, Tastiness, $Self, $Other, Fairness; thresholded at p < 0.05, FWE cluster-level correction, height threshold of p < 0.001) identified VMPFC ([MNI −6, 49, 1], *Figure 3*) as well as a handful of other regions (*Figure 3—figure supplement 1*). This suggests that the VMPFC contains information on trial-wise values of *all* choice-relevant attributes, consistent with its hypothesized importance for valuation and choice.

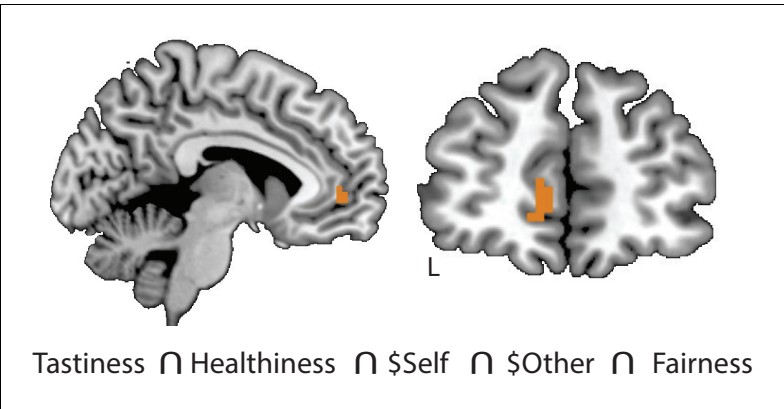

Tastiness ∩ Healthiness ∩ $Self ∩ $Other ∩ Fairness

**Figure 3.** Conjunction of neural representations of attribute values. Multivariate response patterns in the VMPFC encoded trial-wise values of all choice-relevant food attributes (Tastiness, Healthiness) and altruistic attributes ($Self, $Other, Fairness) across regulation conditions, as indicated by a conjunction of attribute-specific decoding maps thresholded at p < 0.05, FWE corrected at cluster-level.

DOI: https://doi.org/10.7554/eLife.31185.009

The following figure supplements are available for figure 3:

**Figure supplement 1.** Conjunction of brain areas that encoded trial-by-trial values of all attributes.

DOI: https://doi.org/10.7554/eLife.31185.010

**Figure supplement 2.** Exploratory functional connectivity analyses.

DOI: https://doi.org/10.7554/eLife.31185.011

**Table 3.** Neural prediction of trial-wise attribute values in food choices and altruistic choices.

| Brain region | Side | T | k | MNI x | y | z |
|---|---|---|---|---|---|---|
| **Main Effect of Healthiness** | | | | | | |
| Dorsolateral Prefrontal Cortex (DLPFC) | L | 5.83 | 24 | −57 | 23 | 34 |
| Lateral PFC (LPFC) | L | 6.29 | 45 | −42 | 35 | 4 |
| LPFC | R | 5.83 | 17 | 54 | 41 | 19 |
| Ventromedial PFC (VMPFC) | R/L | 5.69 | 6 | -3 | 47 | −20 |
| **Main Effect of Tastiness** | | | | | | |
| VMPFC, extends to Mid (MFG) and Superior Frontal Gyrus (SFG) | L/R | 8.29 | 1097 | -9 | 50 | -2 |
| Inferior Parietal Lobe (IPL)/Supramarginal Gyrus (SMG) | R | 6.01 | 39 | 48 | −46 | 46 |
| Pre-Supplemental Motor Area (pre-SMA) | L | 7.06 | 82 | -3 | 23 | 46 |
| SMA | L/R | 6.51 | 100 | 6 | 5 | 70 |
| Motor Cortex | L | 8.85 | 410 | −42 | −28 | 58 |
| Visual Cortex | L | 7.68 | 288 | −30 | −91 | 25 |
| Visual Cortex/IPL/Precuneus | L/R | 7.23 | 1814 | 6 | −61 | 34 |
| Cerebellum | L | 6.23 | 9 | −27 | −70 | −35 |
| **Main Effect of $Self \*** | | | | | | |
| Prefrontal Cortex (VLPFC, DLPFC, VMPFC, DMPFC) | L/R | 5.39 | 1306 | −27 | 50 | 19 |
| SMA | L/R | 4.42 | 111 | 3 | -1 | 55 |
| Visual Cortex | L/R | 6.94 | 2901 | -3 | −82 | 4 |
| **Main Effect of $Other** | | | | | | |
| Dorsomedial PFC (DMPFC) | L/R | 7.16 | 485 | -3 | 44 | 25 |
| VMPFC | R | 5.92 | 108 | 18 | 50 | -2 |
| LPFC | L | 5.58 | 12 | −39 | 32 | 19 |
| Inferior Frontal Gyrus (IFG) | L | 5.51 | 10 | −48 | 26 | -5 |
| SMA | R | 5.54 | 15 | 6 | 23 | 46 |
| Visual cortex | L/R | 7.87 | 661 | -3 | −79 | 4 |
| Cuneus | L | 5.71 | 74 | −24 | −76 | 40 |
| **Main Effect of Fairness** | | | | | | |
| Prefrontal Cortex (includes MPFC, MFG, IFG, right anterior insula) | L/R | 7.54 | 1866 | 45 | 23 | 34 |
| VMPFC | R | 5.84 | 67 | 24 | 59 | 7 |
| Precuneus | L/R | 6.25 | 60 | 0 | −73 | 46 |
| SMG | R | 6.22 | 72 | 60 | −37 | 46 |
| IPL | L | 5.52 | 7 | −39 | −55 | 43 |
| Visual cortex | R | 6.07 | 95 | 12 | −88 | 10 |

Results are reported at a statistical threshold of p < 0.05, FWE corrected at voxel-level (cluster threshold of 5 voxels); * main effect for $Self reported at a statistical threshold of p < 0.05, FWE corrected at cluster-level (height threshold of p < 0.001); only peak activations of clusters are reported; L = left hemisphere, R = right hemisphere, MNI = Montreal Neurological Institute, k = cluster size in voxel

DOI: https://doi.org/10.7554/eLife.31185.008

## Goal-dependent representations of choice attributes and decision values

Having confirmed that attribute values (and decision values) could be decoded from neural response patterns, we next asked whether, how and where neural information content changed as a function of regulatory goals. We hypothesized that altered behavioral weights of an attribute should be mirrored by changes in the neural encoding of that attribute as expressed in varying predictive accuracies. Crucially, these analyses allowed us to test whether goal-dependent change in neural encoding of attribute values occurs in attribute-specific regions or at a common neural locus regardless of

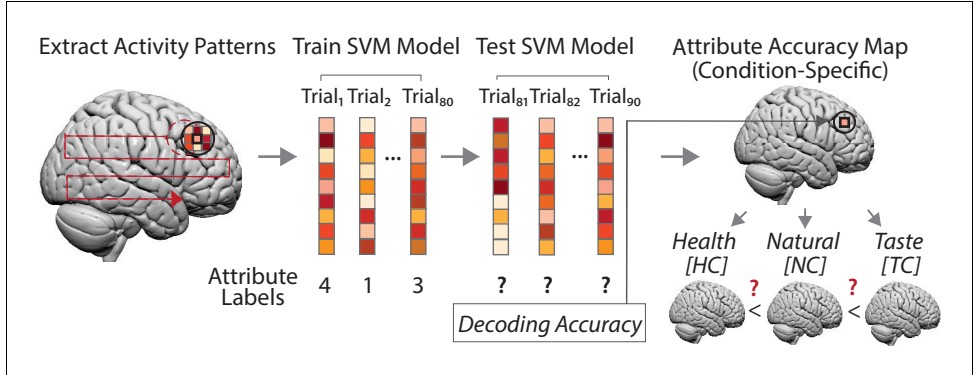

**Figure 4.** Goal-dependent coding of attribute values (left to right). For each participant, we created a spherical searchlight (left panel, black sphere) and extracted multi-voxel response patterns for every trial of a choice task (middle panel). Next, we trained a support vector machine (SVM) regression model with data of 8 runs (80 trials), using neural response patterns as features and trial-wise attribute values as labels (e.g. a food's perceived tastiness). Test data consisted of data of the ninth run (10 trials) for which we predicted the trial-wise attribute values solely based on neural response patterns of these trials. The decoding accuracy (average of 9-fold cross-validation) was assigned to the central voxel of the sphere from which we extracted the neural data (right upper panel). This procedure was repeated for every measured voxel (left panel, dotted red line), yielding a whole brain accuracy map for an attribute, separately for each task condition and participant. Finally, at the group level (lower right panel), we used these whole-brain accuracy maps to test for brain regions where predictive information on an attribute was increased/decreased depending on the task condition, based on predictions of the behavioral computational model (DDM). (Note that condition-specific accuracy maps also allowed testing for main effects of neural encoding of an attribute (i.e. encodes attribute values), irrespective of whether one or several conditions drive the effect.).

DOI: https://doi.org/10.7554/eLife.31185.012

attribute or domain. For each attribute, we used a repeated measures ANOVA implemented in SPM together with condition-specific decoding accuracy maps to test for changes in neural information on attribute values across conditions (see *Figure 4*). This allowed us to identify brain regions where neural information content about an attribute, or decision values (*Supplementary file 1B*), was enhanced or diminished in a way that matched behaviorally-estimated changes in attribute weighting (thresholded at $p < 0.05$, cluster-level corrected, height threshold of $p < 0.001$; see *Table 4*).

## Healthiness
Behavioral model-fitting suggests that healthiness was weighted more heavily in HC compared to both NC and TC (*Figure 2A*). Consistent with model-based predictions, decoding accuracies in the right lateral prefrontal cortex (LPFC) were higher when focusing on health [HC] compared to both other task conditions ([HC >NC], and [HC >TC]) and combined [HC > (NC, TC)]; *Figure 2B*; *Table 4*).

## Tastiness
Behaviorally, tastiness was represented less strongly in HC compared to NC and TC, with no significant differences between the latter (*Figure 2C*). Decoding accuracies in the right superior frontal gyrus (SFG), extending to the mid frontal gyrus (MFG), closely matched these predictions [(NC, TC) > HC] (*Figure 2D*). Neural representations of trial-wise tastiness were also significantly higher for separate comparisons of [NC > HC] and [TC > HC], but did not differ between NC and TC. Only two other regions (visual cortex and left motor cortex) followed this pattern (*Table 4*).

## $Self
Estimates of the best-fitting behavioral parameters for $Self suggest that neural information representing subjects' own benefits should decrease in both pro-social regulation conditions (PC and EC) compared to NC (*Figure 2E*). Formal tests of this pattern ([NC > (EC, PC)]) identified neural responses in both DMPFC (*Figure 2—figure supplement 1*) and the MFG ($p < 0.001$, uncorrected; *Figure 2F*; for [NC] > [EC] significant at $p < 0.05$, cluster-corrected).

**Table 4.** Goal-dependent change of neural information content on attribute values.

| Attribute | Brain region | Side | T | k | MNI | | |
|---|---|---|---|---|---|---|---|
| | | | | | x | y | z |
| **Healthiness** | | | | | | | |
| [HC > (NC, TC)] | (D)LPFC | R | 4.40 | 402 | 51 | 23 | 25 |
| | Visual Cortex | L/R | 6.38 | 593 | 0 | −79 | 7 |
| [HC > NC] | (D)LPFC | R | 4.54 | 241 | 48 | 44 | 19 |
| | Visual Cortex | L/R | 5.52 | 210 | −3 | −79 | 10 |
| [HC > TC] | (D)LPFC | R | 4.28 | 212 | 51 | 23 | 25 |
| | Visual Cortex | L/R | 6.51 | 910 | 3 | −82 | 1 |
| **Tastiness** | | | | | | | |
| [(NC, TC) > HC] | SFG | R | 4.58 | 362 | 24 | 35 | 37 |
| | Motor Cortex | L | 4.46 | 265 | −36 | −16 | 37 |
| | Visual Cortex | L/R | 5.19 | 230 | −3 | −70 | 1 |
| [NC > HC] | SFG | R | 4.08 | 227 | 24 | 35 | 40 |
| | Visual Cortex | L/R | 4.39 | 159 | −3 | −70 | 1 |
| | | R | 4.83 | 252 | 45 | −88 | 14 |
| [TC > HC] | SFG | R | 3.91 | 102 | 24 | 35 | 37 * |
| | Motor Cortex | L | 4.48 | 319 | −48 | −22 | 64 |
| | Visual Cortex | L/R | 4.73 | 123 | −3 | −73 | 1 |
| **$Self** | | | | | | | |
| [NC > (EC, PC)] | DMPFC | L/R | 4.18 | 127 | −12 | 53 | 46 |
| [NC > EC] | DMPFC | L/R | 4.14 | 98 | −3 | 44 | 43 |
| | MFG | R | 3.88 | 52 | 39 | 50 | 34 * |
| **$Other** | | | | | | | |
| [PC > EC] | Precuneus | L | 4.45 | 648 | −15 | −67 | 46 |
| | Temporoparietal junction (TPJ) | R | 3.85 | 170 | 51 | −61 | 16 |
| | Visual cortex | L/R | 4.26 | 276 | −3 | −64 | 4 |
| [(PC, NC) > EC] | Precuneus/TPJ | L/R | 4.70 | 1142 | 12 | −61 | 49 |
| | SMA | L | 4.50 | 189 | −18 | 5 | 67 |
| **Fairness** | | | | | | | |
| [(EC, PC) > NC] | Mid Cingulate Cortex/MFG | L | 5.19 | 118 | −15 | 23 | 31 |
| [PC > NC] | Mid Cingulate Cortex/MFG | L | 5.18 | 183 | −15 | 23 | 31 |

Results are reported at a statistical threshold of p < 0.05, FWE corrected at cluster-level (height threshold of p < 0.001), * indicates clusters that were FDR-corrected at the cluster level; only peak activations of clusters are reported; L = left hemisphere, R = right hemisphere, MNI = Montreal Neurological Institute, k = cluster size in voxels.

DOI: https://doi.org/10.7554/eLife.31185.013

### $Other

Based on the behavioral model we predicted that, compared to NC, the partner's benefits should be represented more strongly when attending to either ethical implications or the other's thoughts and feelings (*Figure 2G*). Surprisingly, no brain regions matched this precise pattern (for [(PC, EC) > NC], or [PC > NC], or [EC > NC], at p < 0.05, cluster-corrected). However, a comparison of [PC > EC] revealed that decoding accuracies in the bilateral precuneus and right temporoparietal junction (TPJ) (*Figure 2H*) (*Figure 2—figure supplement 1*) were significantly more predictive of the others' payoffs when goals focused on the partner compared to ethical implications. Supplemental ROI analyses within these two areas indicated that average predictive accuracies were significantly

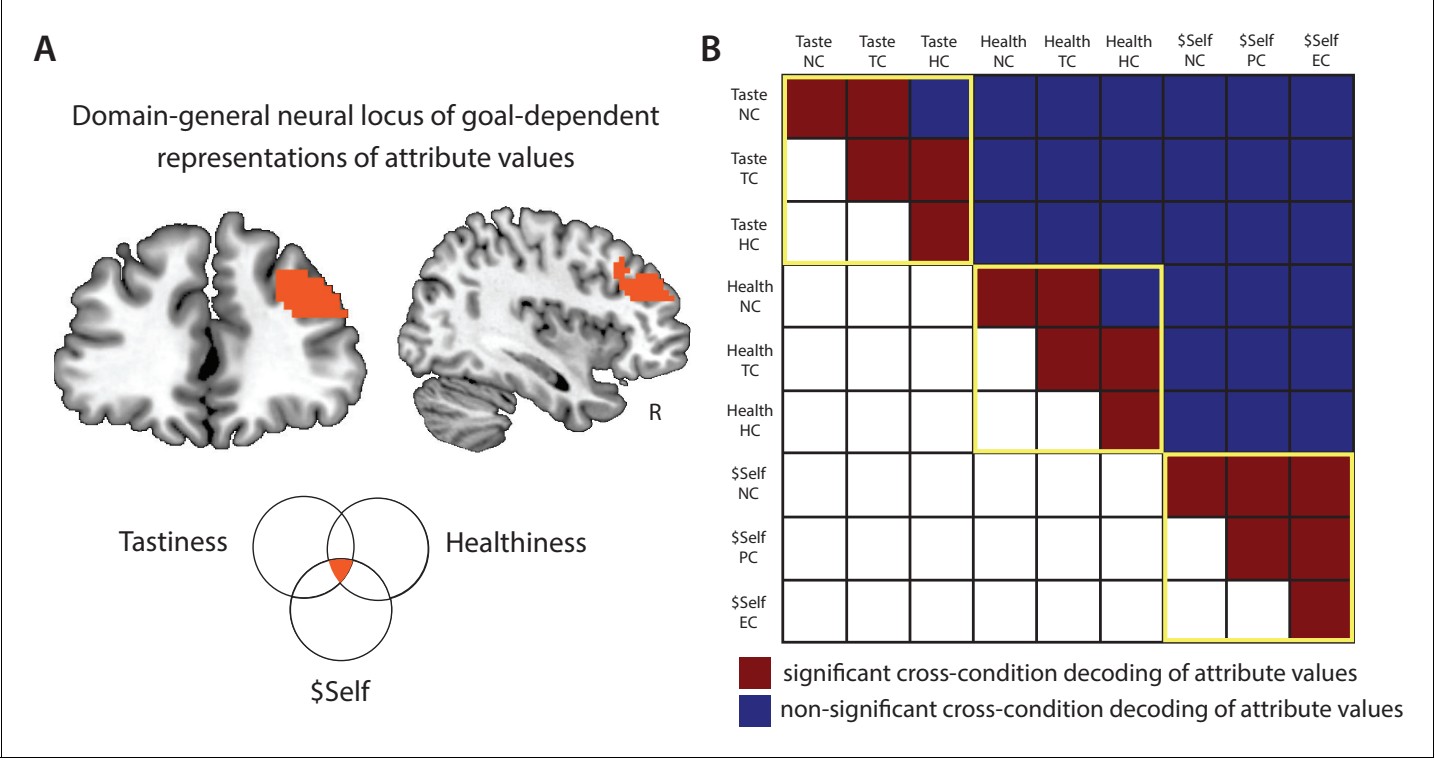

**Figure 5.** Domain-general locus of goal-dependent attribute coding. (**A**) Conjunction of voxels in DLPFC that flexibly encoded attribute values of Healthiness, Tastiness, and $Self across conditions within the respective task (p < 0.05, FWE corrected at cluster-level). (**B**) Cross-condition decoding analyses tested for shared neural code in the DLPFC conjunction area across attributes and regulatory goals. Multivariate SVR models were trained on data in one condition (e.g. Taste NC) and tested on another (e.g. Taste TC), and vice versa (2-fold cross-validation; within-cell sanity checks used split-half approach). Red illustrates significant cross-condition decoding, blue illustrates non-significant results (permutation tests, cutoff-values of 95th percentile of empirical null-distribution). *Within-attribute decoding* (yellow frames): similar neural codes in DLPFC encode values of an attribute across contexts/regulatory conditions (with the exception of 2 of 18 tests). *Cross-attribute decoding*: neural response patterns that encode values of one attribute don't allow predicting values of another attribute (neither within-task [tastiness-healthiness] nor across tasks [tastiness-$Self, healthiness-$Self]), independent of contexts. This pattern of results indicates that goal-sensitive representations of attribute values in DLPFC rely on attribute-specific neural codes.

DOI: https://doi.org/10.7554/eLife.31185.014

higher in PC than NC, partially confirming the prediction of amplified information for $Other [PC > NC] from the behavioral model (*Figure 2H*).

### Fairness

Behaviorally, fairness of payoffs for self and partner influenced choices more strongly when attending to ethics [EC] and, to a lesser extent, the partner's feelings [PC] (*Figure 2I*). Consistent with model-based predictions, decoding accuracies in the left superior frontal sulcus (SFS) predicted the degree of fairness more strongly in the two regulatory conditions compared to natural choice contexts (*Figure 2J*). Contrary to the model prediction, comparisons of [EC > PC] (and [PC > EC]) did not yield any significant results, suggesting that both regulation conditions increased neural representations of fairness considerations to a comparable level.

Notably, repeated measures ANOVAs also allowed testing for changes in neural attribute representations or decision values that were *not* predicted by changes in the behavioral DDM estimates. These supplemental tests did not yield any further significant results (p < 0.05, FWE cluster-corrected).

### Decision values

See *Supplementary file 1B* for details on goal-dependent coding of decision values in both tasks. Only two regions (motor cortex in food task [TC > HC], cerebellum in altruism task [EC >PC]) were

found to be significant (p < 0.05, FWE cluster-corrected). We thus focused on goal-dependent changes in information content on attribute values.

## A common hub for cognitive regulation of attribute values in the DLPFC

To determine whether any areas might serve as a common pathway for goal-dependent changes in encoding of choice attributes, we computed 2-, 3- and 4-way conjunctions of all clusters that showed modulations of predictive information across conditions (*Table 4*). A cluster in the MFG (*Figure 5A*), hereafter referred to as DLPFC, emerged in the 3-way conjunction of voxels that flexibly encoded attribute values for Healthiness, Tastiness, and $Self. We found no other areas showing such a convergence of attributes.

This finding suggests that the DLPFC acts as a domain-general circuit for goal-sensitive value representations. But what does this convergence in the DLPFC signify? On the one hand, the DLPFC might encode a *unitary decision value signal* that is sensitive to current goals. While limited to a specific set of attributes, this would support the integration-level hypothesis. If this was the case, the same code that represents a food's tastiness in the food task (e.g. when focusing on taste) should also permit decoding of other attribute values used in other contexts (i.e., healthiness when focused on health, $Self in natural settings of altruistic choice). On the other hand, the DLPFC might compute attribute-specific representations in a goal-sensitive manner. This hypothesis is more consistent with attribute-level modulation. In this case, encoding of attribute values in this region should be unique to each specific attribute (i.e. codes for one attribute should not permit decoding of other attributes). We tested these competing predictions in a post-hoc ROI-based analysis examining the extent to which neural codes for one attribute in one context (e.g. tastiness in TC) generalize across attributes and contexts (e.g. healthiness in HC). These post-hoc decoding analyses differ from the previous set of analyses: more specifically, to probe for shared neural code in the DLPFC, we trained the SVM regression model on data of one attribute in one condition and see if it allows predicting trial-wise values of *another* attribute in the same or different regulatory condition (and vice versa, 2-fold cross-validation). We also tested for common neural codes for the same attribute across regulatory contexts.

Results most clearly supported the attribute-level hypothesis. While codes for each attribute (tastiness, healthiness, and $Self) in the DLPFC generally allowed for decoding of the same attribute in other conditions at significant or marginally significant levels, no attribute allowed for coding of a *different* attribute, regardless of condition (*Figure 5B*). This supports the idea that the DLPFC acts as a domain-general mechanism for representing different attributes in a goal-sensitive manner, using unique codes for each attribute.

## No evidence for goal-dependent coding of attribute values and decision values in the VMPFC

The vmPFC has previously been suggested to encode attribute values as a function of their current relevance to choice control (*Hare et al., 2011a*). Notably, our analyses on the whole brain level did not reveal any significant variation of attribute value encoding in this area as a function of the regulatory goal. However, in light of previous evidence, we conducted a number of post-hoc ROI-analyses to probe in a more sensitive manner for goal-dependent value coding in the VMPFC (see Appendix 1 – ROI-based post-hoc tests to identify goal-consistent value coding in the VMPFC). While activation patterns in the VMPFC (as well as several other regions) reliably predicted overall decision values in both tasks, regulation failed to modulate decoding accuracies for decision value (*Supplementary file 1C*) or for *any* specific attribute (Appendix 1 – ROI-based post-hoc tests to identify goal-consistent value coding in the VMPFC), and did not predict individual differences in regulatory success (Appendix 1 – ROI-based post-hoc tests to identify goal-consistent value coding in the VMPFC).

## Individual differences in regulatory success

Are some people *generally* more successful using cognitive regulation of decision making or does it depend on the choice domain? Why? To address these questions, we examined the generality and specificity of value representations and their role in regulatory success. In particular, we predicted that if regulatory success operates through common *domain-general* mechanisms, individual success

in regulating the effects of one attribute should be correlated with regulatory success in modifying different attributes in completely different contexts. Consequently, neural responses within such a domain-general neural locus should predict individual differences in people's regulatory success across domains. By contrast, to the extent that cognitive regulation of decision making operates at the attribute-level in a *domain-specific* manner, success regulating one attribute in one domain should be uncorrelated with regulatory success for other attributes in other domains. It should also be predicted by neural activation in distinct, non-overlapping brain regions.

## Regulatory success in goal-dependent attribute weighting

Although our previous analyses suggested that regulatory success as measured by frequency of healthy and generous choices was correlated across participants, this analysis did not examine how such success relates to changes in specific attributes. Thus, to determine whether regulatory success operates through common channels across attributes and domains, we first tested using behavior whether subjects' ability to modulate specific attribute weights (estimated in separate DDMs) was correlated across the two tasks. Consistent with the notion of a common neural mechanism (in DLPFC), successful reduction in the weight on selfish considerations ($\Delta$w $Self) in altruistic choices was correlated with successfully amplifying the weight on health considerations in food choices (e.g., r = 0.50, for $\Delta$w $Self [NC - PC] and $\Delta$w Healthiness [HC - NC], p < 0.05, corrected) and suppressing the weight of taste considerations in food choices (e.g., r = 0.45, $\Delta$w $Self [NC - PC] and $\Delta$w Tastiness [NC - TC], p < 0.05, corrected). Notably, however, enhancement of the weight on another person's outcomes did *not* correlate with changes in other attributes (all p's > 0.05, uncorrected). See *Supplementary file 1C* for detailed list of results. Overall, this pattern suggests that regulation may operate through both common and distinct channels as a function of specific attributes, a point we return to in the neural results below.

## Domain-general predictions of individual differences in regulatory success in DLPFC

Our preceding neural decoding results support a model in which regulation alters specific attribute representations within domain-general brain areas for some attributes (e.g., tastiness, healthiness, $Self) and within domain-specific areas for other attributes (e.g., $Other, fairness). This idea may explain the specific pattern of correlations we observed in behavioral measures of regulatory success and makes a further prediction: if the integrity and flexibility of the DLPFC is only necessary for representing certain attributes in a goal-consistent manner, then responses in this region should predict regulatory success only for those attributes that converge in this area, while regulatory success for other attributes (e.g., $Other) should be predicted by other regions (e.g., TPJ or precuneus). We tested this hypothesis using a cross-subject decoding approach: in a nutshell, this decoding analysis tested whether multi-voxel activation patterns in an ROI (e.g. DLPFC) allowed predicting an individuals regulatory success in a choice task, solely based on the participants regulation-related neural activation patterns (see Materials and methods and Appendix 1 – Multivariate regression of individual differences in regulatory success for details). The analyses focused on an ROI in DLPFC (with supplemental tests for TPJ, precuneus, and VMPFC) and regulatory success scores defined both by changes in attribute weights and by percentage of goal-consistent choices.

As hypothesized, regulation-related neural activation patterns in the right DLPFC conjunction area (*Figure 5A*) during the food task reliably predicted how well a subject decreased taste weights and increased health weights in food choices ($\Delta$w Tastiness [(NC, TC) - HC]: r = 0.51, p < 0.014, permutation test; $\Delta$w Healthiness [HC - (NC, TC)]: r = 0.42, p < 0.041). Predictions further improved when we focused on altered attribute weights for HC versus TC ($\Delta$w Tastiness [TC - HC]: r = 0.68, p = 0.002; $\Delta$w Healthiness [HC - TC]: r = 0.47, p = 0.014). Similar results were found when we predicted subject-specific changes in regulation success based on improved dietary choices ($\Delta$Healthy Choices [HC - (NC, TC)]: r = 0.50, p = 0.016; $\Delta$Healthy Choices [HC - TC]: r = 0.46, p = 0.027), demonstrating that regulation-related neural predictions extend to actual behavior with real consequences.

Next, we asked whether neural activation patterns in the right DLPFC also predict individual differences in regulation success in the altruism task. Remarkably, neural patterns in DLPFC during *food* choices predicted subjects' ability to reduce the weighting of their own monetary payoffs

during *altruistic* choices separated in time by an average of 16 months from the food task (Δw $Self [NC - (EC, PC)]: r = 0.50, p = 0.015; Δw Self [NC - PC]: r = 0.55, p = 0.005; permutation tests). They also predicted increases in generous behavior when attending to pro-social attributes (ΔGenerous Choices [(PC, EC) - NC]: r = 0.63, p < 0.001; ΔGenerous Choices [EC - NC]: r = 0.44, p = 0.028; ΔGenerous Choices [PC - NC]: r = 0.63, p = 0.002). Supplemental analyses suggest that predictive information on altered generosity was driven by neural information on changes in the attribute encoded in the DLPFC ($Self) and not by other attributes of the altruistic choice task (e.g., $Other, fairness) (see Appendix 1 – DLPFC-based prediction of goal-consistent changes of generosity is driven by goal-consistent changes in attribute representations of $Self (but not $Other or Fairness)). We also confirmed that decoding accuracies were not correlated with the delay between both choice tasks (all p's > 0.05, uncorrected), indicating that predictions of individual difference scores of regulatory success were unrelated to temporal delays between tasks. Complementary decoding analyses based on brain data obtained during altruistic choices revealed similar patterns, further supporting our findings (*Supplementary file 1D*).

## Precuneus encodes individual differences in regulatory success in altruistic choice

Strikingly, patterns in the DLPFC did not decode regulatory success for social attributes that were flexibly encoded in other regions of the brain (i.e., $Other, Fairness). A post-hoc analyses tested whether neural activation patterns that encoded values of $Other in a goal-consistent manner would allow predicting individual differences in regulatory success in the altruism task. We found that response patterns in the precuneus reliably predicted individuals' altered generosity in the altruism task (ΔGenerous Choices [PC - EC]: r = 0.57, p = 0.002 [CI: −0.41, 0.38]; ΔGenerous Choices [(NC, PC) - EC]: r = 0.61, p = 0.004 [CI: −0.41, 0.41]), suggesting that domain-specific attribute coding contributes to individual differences in regulatory control.

## VMPFC does not encode individual differences in regulatory success

Because of its hypothesized role in valuation, a post-hoc analyses also examined whether the VMPFC region that encoded all attributes predicted regulatory success in either choice task. However, local activation patterns in VMPFC were *not* predictive of regulatory success for any attribute (all p's > 0.31). This result suggests that while this region may encode all choice-relevant attributes, it was not the locus for changes in value representation in this task. However, exploratory functional connectivity analyses provided subtle hints that the VMPFC could be indirectly related to regulatory success through its modulation of both DLPFC and precuneus (see *Figure 3—figure supplement 2* and Appendix 1 – Changes in functional connectivity with the VMPFC correlate with regulatory success for details).

## Discussion

Cognitive regulation of decision making represents a crucial tool for altering behavior to fit momentary goals (e.g. eat healthy, be kinder). Capitalizing on the strengths of behavioral model-fitting (*Crockett, 2016*) and the greater sensitivity of neural multivariate pattern analysis (*Kriegeskorte et al., 2006*), we demonstrate how regulatory goals modulate value representations at the level of choice-relevant attributes, supporting goal-consistent behavior. Unexpectedly, cognitive regulation of decision making did *not* reliably modulate value signals within the VMPFC. Instead, regulatory effects converged to modulate a subset of distinct attribute representations in both the social and non-social domain within a region of the DLPFC that has previously been implicated in value-based choice (*Hutcherson et al., 2015a*; *Plassmann et al., 2007*; *Plassmann et al., 2010*). Cognitive regulation of decision making also altered attribute representations for specific *social* attributes in distinct areas, including TPJ and precuneus. This pattern of neural convergence and divergence was reflected by behavioral patterns of covariation in regulatory success across tasks, made more remarkable by the fact that they were measured anywhere from weeks to more than a year apart. Our results provide important and novel insights into the domain generality and specificity of cognitive regulation of decision making, explain when and why regulatory success generalizes across contexts and domains, and raise exciting new questions for exploration.

## Attribute-level vs. integration-level effects of cognitive regulation of decision making

Do goals (e.g. eat healthier, be kinder) influence construction of value by operating on distinct attribute representations, or by changing integration of these values in centralized, common-value regions of the brain? Our results provide three key pieces of evidence in favor of attribute-level value modulation by cognitive regulatory control. First, although the VMPFC contained reliable information on the values of *all* attributes and encoded overall decision values across social and non-social contexts, these signals showed *no* modulation by regulatory goal for any attribute or decision value and did not predict individual differences in regulatory success. Moreover, no other area showed a complete correspondence between behavioral and neural effects of regulation, arguing against a single, centralized locus for effects of cognitive regulation on decision making. Second, we observed goal-dependent representations of some attributes (i.e., others' benefits) in distinct, specialized brain regions like the TPJ and precuneus. Third, although we observed converging effects of regulation for a subset of attributes in the DLPFC (including tastiness, healthiness, and self-related benefits), representations of these attributes utilized distinct, differentiated codes. Taken together, although our results do not preclude the possibility that in other contexts cognitive regulation of decision making might operate on a single, centralized value integration mechanism, they suggest that it may often operate by changing distinct attribute representations.

## Domain-general vs. domain-specific effects of cognitive regulation

If cognitive regulation of decision making is mediated by changes in distinct attribute representations, when might we expect regulatory success – or failure – to generalize across contexts and domains? Our results indicate that although the DLPFC used distinct codes to represent different attributes, it may nevertheless be a common denominator in regulatory success across domains. Behaviorally, goal-consistent shifts toward 'virtuous' behavior in one domain (i.e. healthier food choice) correlated with shifts in the other (i.e. more generosity). This covariation was driven by correlated changes in the behavioral weighting of *precisely* those attributes represented in the DLPFC (i. e., tastiness, healthiness, and self-related benefits), but not in attributes encoded elsewhere (i.e. other-related benefits, fairness). These findings are even more remarkable given delays of up to 24 months separating the two choice tasks (average 16 months), ruling out alternative explanations like memory, mood, or priming effects. Thus, the DLPFC may represent a stable individual resource permitting flexible representation of specific attributes according to current goals.

At the same time, goal-consistent changes in pro-social attributes (e.g. others benefits) appeared in areas like the TPJ and precuneus, especially when focused on the partner's thoughts and feelings. This accords with growing evidence linking these regions to *domain-specific* computations related to Theory of Mind (ToM) (*Van Overwalle, 2009*; *Bzdok et al., 2012*; *Schurz et al., 2014*) and representing others' mental states and needs during social choice: for instance, activation patterns in the rTPJ were recently shown to encode individual differences in the level of ToM during altruistic choice (*Tusche et al., 2016*). Notably, activity in these regions did not encode other social attributes (e.g., fairness) or their goal-consistent changes. Moreover, focusing on ethical and normative reasons for giving (which may require less focus on others' specific thoughts and feelings) increased altruistic choice, but actually *decreased* representations of the other's payoffs in these regions. Thus, the TPJ and precuneus appear to encode features specifically related to representing others' outcomes in a goal-sensitive manner, pointing to specialized loci of cognitive regulation in social choice domains.

## The role of VMPFC and DLPFC in valuation and cognitive regulation

Our study adds to a growing body of experimental work finding that behavioral effects of regulation can occur in the absence of corresponding changes to either overall levels of VMPFC response (*Hutcherson et al., 2012*; *Hollmann et al., 2012*; *Yokum and Stice, 2013*), or VMPFC representation of specific attributes like taste (*Hare et al., 2011a*). They also raise the intriguing possibility that the flexibility of DLPFC attribute representations may be particularly important for compensating when regulation of the VMPFC fails, a finding also observed in other studies of cognitive regulation of decision making (*Hutcherson et al., 2012*). This raises an important question: what determines the capacity of the DLPFC to properly represent these different attributes? Intriguingly, exploratory connectivity results suggested that this may actually derive, at least in part, from functional

interactions with the VMPFC area that represented all choice-relevant attributes, with the strength of connectivity between DLPFC and VMPFC correlating with regulatory success. Although speculative, this finding is consistent with research in both animals and humans suggesting that the VMPFC may modulate affective attribute representations in other areas (*Quirk and Beer, 2006*; *Etkin et al., 2006*). These results could also suggest that VMPFC represents an earlier stage in the value construction process, with DLPFC representations emerging more closely to response. Future work including the use of measures with higher temporal precision may help to elucidate when and how interactions between the VMPFC and DLPFC determine regulatory success in different contexts.

## Explaining individual differences in regulatory success and failure

Our study is the first to document goal-consistent changes for *all* choice-relevant attributes, across diverse choice domains, both within and across individuals, shedding light on when and why regulatory efforts may succeed or fail. Our findings point to important divisions in regulatory success as a function of choice attributes and domain: an individual who struggles both to resist cheesecake and ignore their own self-interest may nevertheless have little difficulty in harnessing regulation to represent others' needs and use this as input into social choices. This has important implications in treatment for decision making disorders: if therapeutic interventions fail when focused on one attribute (e.g., be less selfish), a switch to strategies focused on other attributes (e.g., think more about others) might be more effective. Future work will need to explore the full range of domains and attributes in which regulation could play an important role (e.g., risk, intertemporal choice, etc.) in order to determine the extent to which regulatory effects vary or converge across attributes and domains.

It is also worth noting that goal-consistent changes in attribute representations were generally exceptions rather than the rule. *Most* regions permitting attribute decoding showed *no* discernable change in representation of attributes as a function of goal. This may explain why regulatory success often feels so difficult: unregulated attribute representations in some areas (including the VMPFC) may continue to leak into choices, complicating regulatory success. It also argues against a trivial interpretation of our results that the changes we observed are simply uninteresting reflections of behavior: we observed highly specific and localized success-related changes in regions like DLPFC, TPJ, and precuneus, but not in other areas. This suggests that these regions may perform a special role in mediating the impact of regulatory goals on behavior.

## Limitations and future directions

We cannot completely rule out that regulatory affects on behavior and attribute representations might partly reflect differences in motivation to satisfy expectations of the experimenter. However, we note that the specific patterns of convergence and divergence in regulatory success argue against this interpretation of our results: we suspect that if this were the case, we would not have observed either the distinct profile of within-subject correlations in regulatory success for different attributes, or differences in their neural correlates. Nevertheless, further research will be needed to fully resolve the extent to which individual differences in regulatory success result from limits in motivation or limits on capacity. Work examining whether gray matter volume in either the DLPFC and VMPFC predicts regulatory success across individuals might help to resolve such issues (*Schmidt et al., 2018*). Tying laboratory measures of regulation to real-world consequences also remains a necessary future step in understanding the significance of these findings.

Our results also point to a number of other open questions and future directions. The implementation of a strictly data driven approach confirmed that several *a priori* hypothesized regions of interest such as the VMPFC or the DLPFC are crucial for implementing cognitive control of goal-directed choice. However, we cannot rule out that other brain regions not identified by the current analyses (e.g. the ventral striatum) also contribute to decision making during regulation. Indeed, we observed changes in attribute decoding in restricted, non-overlapping areas of visual and motor cortex for some but not all attributes, which might reflect non-causal changes in visual attention or motor preparation, but could also be important precursors to downstream changes in areas like the DLPFC, TPJ and precuneus.

The close correspondence between neural patterns and model-estimated changes in behavioral weighting suggests that our information-based neural measure captured a critical aspect of changes

in neural computations during goal-dependent behavior. However, further investigation is necessary to understand what separates attributes whose representations converged in DLPFC from those that did not. One exciting avenue for future research will be to identify the precise factors that determine whether and when the DLPFC acts as the site for cognitive regulation of value. Understanding this distinction may help to predict when an individual will show more global deficits in regulatory success and when those deficits will tend to stand apart from success or failure in other domains or contexts.

## Materials and methods

### Participants

Fifty-five healthy volunteers (25 female, M ± SD: 28 years ± 5.02) participated in the altruism task. A subset (N = 37, 17 female, 29 years ± 5.24) also completed the food task. Sample size for both established fMRI tasks were selected based on previous successful implementations of the food task (*Hare et al., 2011a*) and the altruism task (*Hutcherson et al., 2015b*). All subjects had normal or corrected-to-normal vision and were free of psychiatric or neurological history. Subjects received $20/hour for their participation, plus the money from a trial selected randomly at the end of the altruism task. They also received a randomly selected food item at the end of the food experiment that had to be consumed in the lab. The altruism data of five subjects and the food data of one subject were excluded from further analyses due to excessive movement (>3 mm/3degree). The altruism data of another subject was excluded from the analysis due to invariant choice behavior. All subjects gave written informed consent and Caltech's Internal Review Board approved the study.

### Tasks

Subjects performed two separate fMRI tasks as part of a large-scale cross-sectional research project. Task order was fixed, with the food task completed on average 16 months (SD: ±8.66; range: 1–24) after the altruism task to specifically probe for common and distinct computations in non-social and social goal-dependent choices.

#### Food task

The non-social fMRI task was a modified version of an established food task (*Hare et al., 2011a*). On every trial, subjects chose between one of 90 food items presented on-screen (4 s) and a default food chosen prior to scanning (*Figure 1A*). Subjects responded by pressing one of four buttons corresponding to 'strong yes', 'yes', 'no', 'strong no' (displayed at the bottom of the screen), using a button box placed in their right hand. The assignment of choice preferences to buttons was fixed throughout the task and the right-left orientation of the scale was counterbalanced across subjects. Inter-trial intervals varied from 1 to 4 s (average of 2 s), during which a white fixation cross was presented against a black background. After scanning, one trial was randomly drawn to determine what the subject would eat before leaving the lab. If subjects failed to respond within the 4 s of the selected trial either the on-screen or the default option was randomly chosen.

Subjects made food choices under three conditions: *Respond Naturally* ('respond as you naturally would', [NC]), *Focus on Health* ('focus on the healthiness of the food when making the choice', [HC]), or *Focus on Taste* ('focus on the tastiness of the food when making the choice', [TC]) (see Appendix 1 – Instructions for regulatory conditions in both choice tasks for instructions). Importantly, subjects were explicitly instructed to always make the decision based on their preference, regardless of the condition. Every condition comprised nine blocks (with 10 trials per block), resulting in a total of 90 trials per condition. Prior to every block, detailed instructions appeared for 4 s. In addition, during food display, a short description ('Respond Naturally', 'Focus on Health', 'Focus on Taste') appeared at the top of the screen to remind participants of the current instruction. Each of the nine functional scanning runs contained one block of every condition (i.e., three task blocks per run), with the order of conditions randomized across runs and subjects. The only exception was the first task block, which was pre-assigned to 'natural' for every subject. Practice trials as well as a short quiz prior to scanning ensured that subjects understood the instructions for each condition and were comfortable with the timing of the task.

Food items varied in their perceived tastiness and healthiness and included healthy snacks (e.g., apples, broccoli) and junk foods (e.g., candy bars, chips). Items were selected based on subjects ratings in a self-paced computerized task prior to scanning that assessed perceived tastiness (5-point Likert scale, 'very untasty' to 'very tasty') and healthiness (5-point Likert scale, 'very unhealthy' to 'very healthy') of 200 food items (*Hare et al., 2011a*; *Hutcherson et al., 2012*). Ninety food items were selected from this larger set to cover the range of health and taste ratings in a roughly uniform manner. In addition, for each subject we chose one default food that was perceived as neutral for taste and health. Each food item was presented once in each of three choice conditions, with presentation order randomized across blocks, functional runs, and subjects. To ensure the motivational saliency of the food items, subjects were asked to refrain from eating 4 hr prior to testing. Stimulus presentation was implemented using high-resolution color pictures (72 dpi) and Psychophysics Toolbox Version 3 (*Brainard, 1997*) together with Matlab (2014a).

## Altruism task

The altruism task was an fMRI compatible version of the dictator game modified from (*Hutcherson et al., 2015b*). On every trial, subjects were presented with a monetary proposal that affected their own ($Self) and another persons' ($Other) monetary payoff (*Figure 1B*). Subjects had 4 s to chose between the on-screen proposal and a constant default allocation ($20 to both) by pressing one of the four response buttons ('strong yes', 'yes', 'no', 'strong no'; direction counter-balanced across subjects). Payouts for self and other ranged from $0 to $40 and always involved a tradeoff between self and other (i.e. prizes for one individual were equal or less than the default, while prizes for the other individual exceeded the default). Thus, subjects always had to choose between acting altruistically (benefitting the other at a cost to oneself) or selfishly (benefitting oneself at a cost to the other) on every trial. At the end of the experiment, one trial was randomly selected and implemented according to the subjects' choice. If subjects failed to respond within 4 s for this trial, both individuals received $0.

Similar to the food task, subjects performed the task under three different conditions: *Respond Naturally* ('respond as you naturally would', [NC]), *Focus on Ethics* ('focus on doing the right thing and consider the ethical or moral implications of your choice', [EC]), or *Focus on Partner* ('focus on your partner's feelings and how the other person is affected by your choice', [PC]). Subjects were reminded to always make their choice based on their preference, regardless of the condition. Conditions were implemented in separate blocks of 10 trials each, with the beginning of a new block signaled by a short reminder instruction (4 s). Matching the food task, subjects performed 9 blocks of each condition (i.e., 90 trials per condition and a total of 270 trials), with the block order counter-balanced across subjects and functional runs, with the exception that the first two blocks were always natural choice trials. Choices in these NC blocks were used to estimate a logistic regression [Choice = $w_{Self}$ * $Self + $w_{Other}$ * $Other] and used for a subject-specific selection of 30% of proposals most likely to elicit generous behavior and 30% of proposals likely to elicit selfish behavior. The remaining 40% of trials were randomly chosen from the full proposal space. Practice trials and a quiz prior to scanning verified that subjects were capable and comfortable to make the choice within 4 s.

## Probabilistic choices

To decrease experimental demand and to ensure anonymity in the altruism task, subjects were informed that implementation of their choices was probabilistic and that in 40% of trials their choices would be reversed (*Hutcherson et al., 2015b*). Subjects were informed that their partner would only know the proposal and the outcome of the randomly chosen trial, but not their decision (i.e., if the outcome was due to the subjects' choice or a choice reversal). The implementation was as follows: After each choice (jittered delay of 2–4 s), an outcome screen (4 s) informed subjects of the implementation of choices (implemented/choice reversal), followed by a jittered inter-trial interval of 1–4 s (average of 2 s) before the next choice screen appeared. Computerized control questions during training confirmed that subjects understood the probabilistic nature of the task and that it was still in their best interest to choose according to their individual preferences. In the food task, we matched the probabilistic implantation in the altruism task, and informed participants prior to scanning that their choices would be implemented with 60% probability.

Data from an independent behavioral pilot study (N = 17, 11 female, M ± SD: 24.12 years ± 5.83) confirmed that choices under almost perfect implementation (90%) closely matched those observed under 60% implementation conditions (within-subject design, all p's > 0.37, uncorrected, for paired t-tests of RTs, percentage of generous and healthy choices). These findings strongly suggest that the probabilistic nature of the task did not systematically alter preference-based choices in both tasks.

## Behavioral computational model (DDM)

We used a multi-attribute extension of the standard drift diffusion model (DDM) (*Ratcliff and McKoon, 2008*; *Smith and Ratcliff, 2004*) to capture behavior in both the food and altruism task, using a maximum-likelihood procedure similar to that described in (*Hutcherson et al., 2015b*) to find the best-fitting parameters (see Appendix 1 – Drift diffusion model for details). For capturing behavior in the food task, we fit a model using five parameters: two parameters for the weights on tastiness and healthiness, a parameter for non-decision time (NDT) representing perceptual and motor processes, and two parameters specifying the initial height of the choice-determining threshold (b) as well as the exponential decay rate of this threshold toward zero (d) as the time limit for responding approached. For capturing behavior in the altruism task, we fit a model using six parameters: three parameters related to the weights on $Self, $Other, and fairness ($-1*|$Self - $Other|$), as well as parameters related to NDT, b, and d (see *Supplementary file 1A* for details).

## Functional image acquisition

Functional imaging was performed on a 3T MRI scanner (Magnetom Trio, Tim System, Siemens Medical Systems, Erlangen) equipped with a 32-channel head coil. T2*-weighted functional images were obtained using an echoplanar imaging (EPI) sequence (TR = 2.5 s, TE = 30 ms, flip angle = 85°, $3 \times 3 \times 3$ mm, matrix size $64 \times 64$, 47 axial slices, descending sequential acquisition order). For the altruism task, a maximum of 1521 volumes were acquired. For the food task we acquired 990 volumes. High-resolution T1-weighted structural images were acquired at the end of each scanning session using an MPRAGE sequence (TR = 1.5 s, TE = 2.91 ms, flip angle = 10°, TI = 800 ms, $1 \times 1 \times 1$ mm, matrix size $256 \times 256$, 176 slices).

## fMRI data analysis

Functional images were analyzed using the statistical parametric mapping software SPM12 (http://www.fil.ion.ucl.ac.uk/spm) implemented in Matlab. Preprocessing consisted of slice-time correction (reference slice 47), spatial realignment (by first registering each subjects' data to the first image of each run, then all functional runs were co-registered with each other), and normalization to the Montreal Neurological Institute (MNI) brain template (EPI template). For every subject, we estimated several general linear models (GLMs), using a canonical hemodynamic response function (hrf), and a 128 s high-pass cutoff filter to eliminate low-frequency drifts in the data.

### Trial-wise estimates of choice phases: GLM1 (food task) and GLM2 (altruism task)

These GLMs aimed to identify brain responses that encode trial-by-trial variations in attributes (i.e., foods' healthiness or tastiness in the food task; payoffs for subjects and confederate and the fairness of the offer in the altruism task) and decision-values (four-point response from 'strong no' to 'strong yes') during choice periods. To this end, these models obtained a trial-wise measure of BOLD responses during food (GLM1) and altruistic choices (GLM2) at the time of the choice. For each subject, GLM1 included a regressor for each choice period (R1-R270) in the food task, lasting from the onset of a food presentation to the button press that represented the choice for the trial. In addition, the model estimated a separate regressor for the outcome phases for each functional run, movement parameters, and run-wise session constants as regressors of no interest. GLM2 mirrored GLM1 and estimated regressors of interest for every altruistic choice (R1-R270), lasting from the onset of the monetary proposal to the button press that signified the choice in this trial. GLM2 also estimated regressors of no interest including outcome phases, movement parameters, and session constants. Estimated responses for the regressors of interest – the choice periods of each task (R1-

R270 from GLM1 and 2, respectively) – were then used as inputs for the multivariate decoding analyses (support vector regressions, SVRs) described below.

## Neural computational model: within-subject decoding of choice attributes

This multivariate pattern analysis (MVPA) aimed to identify brain regions that encode trial-by-trial fluctuations of choice-relevant attributes (e.g. foods healthiness, payoff to self) or decision values (four-point response from 'strong no' to 'strong yes'), and to assess how current goals affect neural information on the attribute level. Thus, these decoding analyses allowed us to explicitly test if regulation-based changes in *neural* information on choice-relevant variables (e.g., healthiness of foods) matched predictions from the *behavioral* computational model.

For each choice attribute and each condition, we applied a separate support vector regression (SVR) analysis in combination with a whole-brain 'searchlight' approach (*Figure 4*). The key advantage of the searchlight decoding approach is that it does not depend on a priori assumptions about informative brain regions and ensures unbiased information mapping throughout the whole brain (*Kriegeskorte et al., 2006*; *Haynes et al., 2007*). For every subject, we defined a sphere with a radius of 4 voxels around a given voxel $v_i$ of the measured brain volume (*Tusche et al., 2016*; *Wisniewski et al., 2015*; *Kahnt et al., 2011*; *Heinzle et al., 2012*) For each of the N voxels within this sphere, we extracted trial-wise parameter estimates of a particular condition (i.e., 90 of the 270 trial-wise regressors of choice periods from GLM1 (food task) or GLM2 (altruism task)). N-dimensional pattern vectors were created separately for each of the 90 trials of the respective fMRI task. Neural pattern vectors for 8 of the 9 task blocks ('training data') served as input features, with trial-wise values of the attribute (e.g., healthiness rating) as labels of the prediction. The prediction was realized using a linear kernel support vector machine regression (http://www.csie.ntu.edu.tw/~cjlin/libsvm) (v-SVR) with a fixed cost parameter c = 0.01 that was preselected based on previous implementations of this decoding approach (*Tusche et al., 2016*; *Kahnt et al., 2011*; *Kahnt et al., 2014*; *Gross et al., 2014*). The resulting model provided the basis for the prediction of the trial-wise values of an attribute (e.g. healthiness ratings) of the 10 trials of the remaining task block ('test data') based on their neural response patterns. This procedure was repeated nine times, always using pattern vectors of a different task block as test data, yielding a 9-fold cross-validation. Predictive information about the choice attribute was defined as the average Fisher's z-transformed correlation coefficient between the value predicted by the SVR model and the actual values of an attribute in these trials (*Tusche et al., 2016*; *Kahnt et al., 2011*; *Kahnt et al., 2014*; *Gross et al., 2014*). This decoding accuracy value was assigned to the central voxel of the searchlight. The procedure was repeated for every voxel of the measured brain volume, yielding a three-dimensional decoding accuracy map for every subject, separately for each choice attribute and each condition. Decoding maps were smoothed (6 mm full width at half maximum, FWHM) and submitted to two different random-effects group analyses.

First, to establish that neural response patterns encode the current value of a choice-relevant attribute during choices, we averaged subjects' decoding accuracy maps for a particular attribute obtained in the three conditions (e.g., separate SVRs for healthiness in NC, HC, and TC). Subject-specific average information maps were than used in a random effect second level analysis (single t-test as implemented in SPM) and tested against chance level at a statistical threshold of p < 0.05 (FWE cluster-corrected, height threshold of p < 0.001). Note that if resulting cluster sizes at this statistical threshold prevented effective functional localization (i.e., clusters that exceeded 6000 voxels), we report results at p < 0.05, FWE corrected at voxel-level. Second, to examine if predictive neural information on a choice attribute varied systematically across the three conditions of the respective task, we used a repeated measures ANOVA as implemented in SPM. Only regions that passed the statistical threshold of p < 0.05 (FWE corrected at cluster-level, height threshold of p < 0.001) are reported.

We also compared the results of the multivariate SVRs with those of a conventional univariate analysis (see *Supplementary file 1E*, and Appendix 1 – Univariate Analysis of fMRI Data). However, note that multivariate decoding approaches have been prosed to be more sensitive than traditional mass-univariate approaches: because multivariate pattern classifiers take advantage of information encoded across multiple voxels and exploit systematic differences in voxel selectivity within a

specific brain region, they have been suggested to detect information that would be missed by conventional analyses (*Kriegeskorte and Bandettini, 2007*).

There is a potential concern that the some intervals of the response scales for choice attributes (e.g. tastiness) or decision values ('strong no', 'no', 'yes', 'strong yes') might be subjectively larger than other intervals. While the present data don't allow ruling out this potential concern, previous implementations of the response scales in both established tasks suggest that the operationalization of attribute-specific judgments and decision values allows to reliably identify value signals in the brain (*Hare et al., 2011a*; *Hutcherson et al., 2012*; *Hare et al., 2009*; *Hutcherson et al., 2015b*).

## Neural computational model: cross-subject decoding of individual differences in regulatory success

This multivariate decoding analysis investigated whether neural activation patterns predict individual differences in regulatory success. A cross-subject decoding approach was used to test for information on the *degree* to which cognitive regulation affected attribute weights and choices. Clusters identified in the repeated measures ANOVA (see above) were defined as regions of interest (ROIs). Importantly, a main goal of our study was to test for potential common neural substrates underlying context-sensitive weighting of choice-attributes across choice-domains. Hence, these decoding analyses focused on voxels identified in a formal conjunction of the significant clusters for flexible representations of Healthiness, Tastiness, and $Self (at p < 0.05, FWE corrected at cluster-level, height threshold of p < 0.001; see *Figure 5A*). First, we tested if neural activation in this ROI obtained during food choices encodes subject-specific regulation success in the food task, but also in an independent social altruism task. To this end, we extracted parameter estimates for all voxels in the ROI (*Figure 5A*) from subjects first-level GLM1 (food choices) using the contrast image [HC > (NC, TC)] (based on DDM results suggesting differential attribute representations for this comparison, *Figure 2A,C*). Resulting pattern vectors (one per subject) were used as input features for the prediction and individual difference scores in regulation success served as labels. Regulation success was defined using difference scores in DDM parameters (e.g., $\Delta$w Tastiness [HC − (NC, TC)]) and in observed choice behavior (e.g., $\Delta$Healthy choices [HC − (NC, TC)]). Predictions used a linear $\nu$-SVR (libSVM) with a fixed cost parameter c = 0.01 (similar to within-subject decoding) and a leave-one-subject out approach (yielding a 36-fold cross-validation). Decoding accuracies reflect correlations of the observed and predicted regulation score. Statistical significance was assessed by comparisons to empirical null-distribution (realized by randomly permuting the pairing of subjects' neural pattern vectors and behavioral regulation scores 1000 times). Only decoding accuracies above the 95th percentile of null-distributions were considered statistically significant (*Corradi-Dell'Acqua et al., 2016*). As a sanity check, analyses were repeated training on data obtained during altruistic choices (see *Supplementary file 1D*).

Note that *ROI-based cross-subject* decoding analyses used permutation tests to assess the statistical significance of the predictions instead of simple t-tests (as implemented in SPM) applied to the *whole-brain within-subject* searchlight decoding maps. Regarding the latter, computational costs for estimating empirical null-distributions for several tens of thousands of searchlight analyses - implemented separately for every attribute, decision-value, condition, and subject - prevented us from using permutation tests for whole-brain decoding analyses. However, supplemental analyses that used t-tests to statistically assess ROI-based results yielded similar results as permutation tests, demonstrating that both statistical approaches generate comparable interpretations for the present data. Notably, empirical permutation-based null-distributions for ROIs also confirmed the theoretical chance level of the prediction that underlies statistical inferences for the whole-brain searchlight results (t-tests as implemented in SPM). Nevertheless, statistical tests based on empirical null-distributions can be viewed as superior insofar as they address the potential concern that means and distribution of predictions based on chance alone might vary across brain regions.

Note also that the ROI (*Figure 5A*) used for this analysis was defined based on a fully independent and orthogonal set of tests for altered decoding accuracies across task conditions at the group level. Thus, ROI-selection was not subject to double dipping (*Kriegeskorte et al., 2009*).

## Acknowledgements

This research was supported by funding from NIMH Conte Center 2P50 MH094258. We also thank Ralph Adolphs and Antonio Rangel for their support for the project.

## Additional information

### Funding

| Funder | Grant reference number | Author |
| --- | --- | --- |
| National Institute of Mental Health | NIMH Conte Center 2P50 MH094258 | Cendri A Hutcherson |

This research was supported by funding from NIMH Conte Center 2P50 MH094258. The funders had no role in study design, data collection and interpretation, or the decision to submit the work for publication.

### Author contributions

Anita Tusche, Cendri A Hutcherson, Conceptualization, Data curation, Software, Formal analysis, Validation, Investigation, Visualization, Methodology, Writing—original draft, Project administration, Writing—review and editing, designed the study, collected the data, performed data analysis, and wrote the manuscript

### Author ORCIDs

Anita Tusche (iD) http://orcid.org/0000-0003-4180-8447

### Ethics

Human subjects: All subjects gave written informed consent, and consent to publish, and the Internal Review Board of the California Institute of Technology approved the study (AR-346).

### Decision letter and Author response

Decision letter https://doi.org/10.7554/eLife.31185.029
Author response https://doi.org/10.7554/eLife.31185.030

## Additional files

### Supplementary files

• Supplementary file 1. Supplemental tables. (**A**) Drift diffusion model (DDM) parameters and RTs. Table reports Mean and Standard Deviations (±SD). b = initial height of the barrier; d = rate of decay for collapsing barriers; NDT = non decision time.; RTs = reaction times. Results for the Altruism Task are reported both for full sample (N = 49), and for the subset of participants that also completing the Food Task (N = 36).

(**B**) Support vector regressions (SVRs) of decision values (DV) in both choice tasks. Results for goal-independent encoding of DVs (Main effect for DV, averaged information content across three conditions in each task) are reported at a statistical threshold of $p < 0.05$, FWE corrected at voxel-level; [†] indicates results for goal-dependent variations in DV (repeated measures ANOVAs across three conditions in each task) reported at $p < 0.001$, FWE cluster-corrected at $p < 0.05$, k = 10 voxels; only peak activations of clusters are reported; L = left hemisphere, R = right hemisphere, k = cluster size in voxels, MNI = Montreal Neurological Institute.

(**C**) Correlation of regulatory success in goal-dependent attribute weighting across choice domains. Correlation coefficients for change scores in attribute weights (w) in food choices and altruistic choices estimated in two separate DDMs. Successful reduction in the weight on selfish considerations (Δw \$Self) in altruistic choices was correlated with successfully amplifying the weight on health considerations (Δw Healthiness) suppressing the weight of taste considerations (Δw Tastiness) in food choices. Changes in the ability to increase social considerations (Δw Other) were not correlated

to goal-consistent changes in food attributes. Δw Tastiness [NC - TC] is not displayed, as estimated attribute weights did not significantly differ between conditions. Note also that differences scores in Δw Healthiness [NC - TC] (last column) were minimal, limiting the interpretability of the respective correlation analyses.

(**D**) Decoding of individual differences in regulatory success in DLPFC (altruism task). Decoding of individual differences in regulation success based on response patterns in right DLPFC (*Figure 5A*) obtained in the altruism task. Response patterns reliably predicted the extent of increased generous choice behavior. Consistent with key results reported in the main text, neural activation patterns also predicted individual's increased healthy choices in a separate food task. Regarding altered attributes weights, predictive information in DLPFC was selective for subjects' inhibition of $Self weights, but did not extend to altered weights on $Other or Fairness, confirming results reported in the main text. Higher-than-chance predictions are reported when decoding accuracy values exceeded the 95th percentile of empirical null-distribution (cutoff), obtained with 1000 replications of the analysis on permuted data sets.

(**E**) Univariate encoding of attributes in food task and altruism task. Regions reported as significant if they passed a cluster-corrected threshold p < 0.05, with a voxel-defining threshold of p < 0.001, uncorrected, unless otherwise noted. * Illustrates results significant at p < 0.001, uncorrected, reported for completeness: subgenual area did not overlap with the area of vmPFC that displayed overlapping representations of all attributes; only peak activations of clusters are reported; L = left hemisphere, R = right hemisphere, MNI = Montreal Neurological Institute, k = cluster size in voxels.
DOI: https://doi.org/10.7554/eLife.31185.015

• Transparent reporting form
DOI: https://doi.org/10.7554/eLife.31185.016

## Data availability

Functional imaging and behavioral data is deposited at the project's Open Science Framework (OSF) page (osf.io/wa4cs). The project page also makes available the derived statistical maps (univariate and multivariate decoding analyses), regions of interest (ROIs) used in analyses of functional imaging data, processed behavioural data, and details on the experimental procedure.

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

## Appendix 1

DOI: https://doi.org/10.7554/eLife.31185.017

### Instructions for regulatory conditions in both choice tasks

#### Food task

- *Focus on Health [HC].* On some trials, when you make decisions, we would like you to focus on the HEALTHINESS OF THE FOOD. In other words, you should think about the how healthy the food is (i.e., is it nutritious, good for you, etc.?). Try to bring your actions in line with these considerations. We will indicate these trials with the instruction 'FOCUS ON HEALTH.'
- *Focus on Taste [TC].* On other trials, when you make decisions, we would like you to focus on TASTINESS OF THE FOOD. In other words, you should think about the food tastes (i.e., is it delicious, savory, etc.?). Try to bring your actions in line with these considerations. We will indicate these trials with the instruction 'FOCUS ON TASTE.'
- *Respond Naturally [NC].* Finally, on a third type of trial, when you make decisions, we would like you to JUST CHOOSE HOW YOU NATURALLY WOULD. In other words, you should allow whatever thoughts and feelings come most naturally to you. We will indicate these trials with the instruction 'RESPOND NATURALLY.'

#### Altruism task

- *Focus on Ethics [EC].* On some trials, when you make a decision, we would like you to focus on DOING THE RIGHT THING. In other words, you should think about the justice of your choice and its ethical or moral implications. Is the choice you are making the moral thing to do? Try to bring your actions in line with these considerations. We will indicate these trials with the instruction 'FOCUS ON ETHICS.'
- *Focus on Partner [PC].* On some trials, when you make a decision, we would like you to focus on YOUR PARTNER'S FFELINGS. In other words, you should think about the other person affected by your choices and how much they would like the money. Will they be happy with your choice? Try to bring your actions in line with these considerations. We will indicate these trials with the instruction 'FOCUS ON PARTNER.'
- *Respond Naturally [NC].* On some trials, when you make a decision, we would like you to JUST CHOOSE HOW YOU NATURALLY WOULD. In other words, you should allow whatever thoughts and feelings come most naturally to you. We will indicate these trials with the instruction 'RESPOND NATURALLY.'

### Drift diffusion model

On every trial the model assumes that a choice results from a dynamically evolving stochastic relative decision value (RDV) signal that provides an estimate of the desirability of the proposed prize. In the case of foods, this consisted of a combination of two attributes (health and taste; as rated by the subject outside of the scanner). For altruistic choices this consisted of three attributes ($Self, $Other, and Fairness defined by [$-1*$|$Self - $Other|]; based on monetary values in the trials proposal). On each trial, the RDV signal starts at zero and, after an initial non-decision time (NDT) related to perception or memory processes, accumulates stochastically at time *t* according to *Equation 1* (food choices) and *Equation 2* (altruistic choices):

$$RDV_t = RDV_{t-1} + w_{taste}(Taste - Default) + w_{health}(Health - Default) + \varepsilon_t \quad (1)$$

$$RDV_t = RDV_{t-1} + w_{self}(\$Self - \$20) + w_{other}(\$Other - \$20) + w_{ineq}(Fairness) + \varepsilon_t \quad (2)$$

Weights (w) are constant across time and trials, and $\varepsilon_t$ denotes white Gaussian noise that is identically and independently distributed with standard deviation $\sigma$. Consistent with previous work, we fixed $\sigma = 0.1$, and measure other parameters in proportion to $\sigma$ (Ratcliff and McKoon, 2008). The model assumes that the on-screen option is accepted if the positive barrier is crossed first and rejected if the negative barrier is crossed first. Reaction time (RT) equals the sum of the NDT and crossing time $t$. As in our previous work (Hutcherson et al., 2015) we allowed for the possibility of collapsing barriers. The barriers are symmetric and described by the equation

$$\pm B_t = be^{-td}$$

where $b$ denotes the initial height of the barrier, $d \geq 0$ denotes the rate of decay, and $t$ is measured from the end of the non-decision period. The models for both choice tasks thus have free parameters related to the weights on relevant attributes as well as *NDT*, $b$ and $d$.

Model fitting was performed similarly to our previous work (**Hutcherson et al., 2015a**), with some important modifications to reduce computational time. In brief, we began with a range of discrete possible values for each parameter, which defined an N-dimensional grid for the full combination of all N parameters. To measure attributes for food and altruistic choices on a common scale, food attribute ratings were converted into the same range as monetary proposals for self and other (i.e., −20 to +20), using the linear transformation

$$Rating = 5 * Rating - -15.$$

This allowed us to explore a common set of parameter values for the two tasks: $w_{health} = w_{taste} = w_{self} = w_{other} = w_{fairness} = [-0.012, -0.01, \ldots -0.002, -0.001, 0, 0.001, 0.002, 0.004, \ldots.024]$, **NDT** = [0.25, 0.5, … 1.25], **b** = [0.1, 0.11, 0.12, …. 38, 0.39, 0.4] and **d** = [0, 0.00025, 0.0005, 0.001, 0.00125, 0.0015, 0.00175, 0.002]. We determined all unique RDVs resulting from the combination of trial-level attribute values (e.g., \$Self, \$Other, and Fairness; or Taste and Health) and distinct attribute weights, and then simulated distributions of choices and RTs for all unique combinations of values, barrier parameters, and non-decision times.

In addition, to speed up computation time while generating more complete simulations of choice and RT, we used an approximation of the DDM in which we divided the space between the two thresholds into 50 contiguous bins representing possible locations of the accumulated RDV signal. Next, we specified a transition matrix representing the probability of transitioning from bin $i$ to bin $j$, given a specified mean drift rate. This was calculated by assuming that the transition probability could be described as a Normal distribution with mean = RDV, $SD = \sigma\sqrt{.001}$. In other words, the likelihood of transitioning from bin $i$ to bin $j$ depends on the relative positions of their centers, with transitions of size and direction RDV being more likely. To compute the probability density function representing the possible locations of the DDM at time $t$, we multiplied a vector representing the current probability density of DDM locations over all 50 bins by the transition matrix. We assumed that at time $t = 0$, 100% of the DDMs started at the point halfway between the two thresholds. In addition, we assumed that the two barriers represent absorbing states, such that the probability of transitioning out of bins with centers beyond $\pm B_t$ is equal to 0. To compute the probability of a DDM terminating at time $t$, we subtracted the cumulative probabilities of evidence having accumulated past $\pm B_t$ at time $t$ from that same quantity at time t+1. Although this method is an approximation to the DDM, it is consistent with recent approaches to fitting drift diffusion models to behavioral evidence (**Brunton et al., 2013**), and allows us to compute a smoother estimate of the probability of observing a particular choice at time $t$. Our previous work using individual simulations required us to compute the likelihood of an RT falling into 250ms bins. The current approach allows us to compute this probability for each millisecond from choice onset to the response time limit for a given set of parameters. Matlab code for this approximation is available upon request.

We used these simulated likelihoods to estimate the best-fitting combination of parameter values, separately for each subject in each condition, using a modified version of maximum likelihood estimation. Based on the simulated likelihood distributions, we calculated the likelihood of the observed combinations of choice and RT on every trial for every subject under a given combination of parameters. We then identified the set of parameters that

minimized the negative log likelihood of the selected trials in a two-step process. To minimize exploding computational costs, we first performed a grid search to identify the combination of parameter values that maximized the data likelihood over a courser combination of parameter values. We then performed a finer grid search on a range of parameter values centered on these values, in order to estimate parameters at a finer resolution. Although this strategy has the disadvantage that it may miss the true minimum among all $1.29*10e^2$ *Shaw et al., 2005* combinations, it substantially reduces computational costs, provides satisfactory fits to the data, and is consistent with previous approaches (*Krajbich et al., 2010*).

One concern is that regulation may introduce additional cognitive complexity into a task that makes the DDM ill suited to capture behavior. To address this concern, and to determine whether the DDM models were able to accurately capture differences in behavior across conditions, we computed the mean squared error (MSE) between the predicted likelihood of accepting a proposal and the observed choice, separately for each subject in each condition of the altruism task. These analyses suggested that the DDM was slightly but significantly *better* at predicting choices in the two regulation conditions (mean $MSE_{PC} = 0.108 \pm 0.052$, mean $MSE_{EC} = 0.102 \pm 0.052$, mean $MSE_{NC} = 0.120 \pm 0.043$, both p's < 0.05), but that the two regulation conditions did not differ from each other (p = 0.32). We performed a similar analysis examining MSE for behavior in the food task. We observed no difference in predictive accuracy between the Natural and Health conditions ($MSE_{NC} = 0.100 \pm 0.050$, $MSE_{HC} = 0.102 \pm 0.060$, p = 0.31), and observed a small but significant improvement in predictive accuracy for the Taste condition ($MSE_{TC} = 0.088 \pm 0.045$). These results suggest that the DDM provided a reasonable description of behavior in all three conditions of tasks.

## ROI-based post-hoc tests to identify goal-consistent value coding in the VMPFC

We explicitly tested whether the VMPFC encodes attribute values as a function of their current relevance to choice. Notably, our analyses on the whole brain level did not reveal any significant variation of attribute value encoding in this area as a function of the regulatory goal. However, in light of previous evidence, we conducted a number of post-hoc ROI-analyses to probe in a more sensitive manner for goal-dependent value coding in the VMPFC. The ROI consisted of voxels in the VMPFC identified in the conjunction analyses (see *Figure 3* for illustration). We tested this question in three different sets of analyses:

1. For the first approach, we examined average predictive information in the VMPFC-ROI identified by the whole brain searchlight decoding analyses described in the paper: For each participant and each condition, we extracted decoding accuracies for each voxel in the VMPFC-ROI and estimated an ROI-based average for each choice-attribute. Next, for each attribute, we ran a repeated-measures ANOVA to explicitly test for altered information content. Confirming results from whole-brain analyses, average neural information about the values of choice attributes (as measured by decoding accuracies) was not significantly modulated by the regulatory goal (all p's > 0.13, uncorrected). This finding is in line with our whole-brain results (implemented in SPM), and suggests that the lack of results was not due to overly stringent statistical thresholds. This provides further support for our interpretation that value representations in the VMPFC on the attribute level are unaffected at a group level by our implemented regulatory goals.

2. However, this analysis leaves open the possibility that local response patterns in the VMPFC (rather than average decoding accuracies) might differentiate attributes by regulatory goal. Thus, we ran supplemental within-subject decoding analyses to predict trial-wise values of an attribute based on local response patterns in the VMPFC. These analyses were similar to the searchlight decoding approach with the difference that ROI-wise response patterns in VMPFC served as neural features for the predictions (i.e., instead of extracting multi-voxel patterns from a sphere around individual voxels in the ROI, we use one response pattern consisting of all voxels of the ROI). For each attribute separately, we estimated how well a subject's multi-voxel response patterns encoded attribute values in a particular task condition. Next, on a group level, we used repeated measures ANOVA to examine whether neural

information on an attribute varied depending on the behavioral goal. Consistent with the searchlight decoding approach, predictive information encoded in the VMPFC did not significantly vary cross task conditions (all p's > 0.29).

3. Finally, we examined if response patterns in the VMPFC encoded goal-dependent changes *across* participants. Such a pattern might suggest that even though we did not observe main effects, VMPFC attribute representations still contribute in some way to regulatory success. Thus, in addition to the within-subject analyses described above (1-2), we also ran cross-subject decoding analyses with an approach that was identical to the decoding analyses of individual regulatory success from neural response patterns of the DLPFC, with the exception that response patterns were extracted from the VMPFC cluster identified in the conjunction analysis. Consistent with the lack of main effects for the supplemental analyses reported above, cross-subject decoding analyses did not yield significant predictions of participants' regulatory success for any attribute or for choice behavior (i.e. healthy or generous choices) in the food task or altruism task (all p's > 0.31).

To address potential concerns that the selected ROI might have been too small for the multi-voxel analyses approach, we also repeated these analyses (1-3) using a slightly larger ROI. Results of the analyses (1-3) matched those for the original VMPFC ROI, suggesting that null-findings in supplemental analyses are not merely driven by the comparatively small number of voxels identified in the conjunction analyses (Tastiness, Healthiness, $Self, $ Other, Fairness).

Taken together, these more sensitive and fine-grained ROI analyses confirm our conclusion that, in this task, our VMPFC ROI encodes the values of each of the individual attributes, but not in a manner that is sensitive to regulatory goals.

## Multivariate regression of individual differences in regulatory success

We used a multivariate cross-subject decoding approach to examine whether neural activation patterns in the DLPFC (*Figure 5A*) predict individual differences in the *degree* to which regulatory manipulations affected attribute weights and choices. Results for this decoding analysis reported in the main text refer to an analysis based on neural responses in the DLPFC obtained during *food* choices (extracted from subject-specific contrast images for [HC > (NC, TC)]). We found that these neural activation patterns predict individual differences in regulatory success not only for the food task, but also for regulation success in a completely independent altruism task.

For the sake of completeness, we repeated this analysis using DLPFC response patterns obtained during *altruistic* choices. The cross-subject decoding approach mirrored the one for the analysis described in the main text: First, we tested if neural activation in the DLPFC obtained during *altruistic* choices encodes subject-specific regulation success in the *altruism* task. Next, we tested if predictive information would generalize to a completely independent (non-social) food task separated in time by an average of 16 months. To this end, we extracted parameter estimates for all voxels in the ROI (*Figure 5A*, similar to the ROI used for analysis based on food choice data) from participants first-level GLM2 (altruism task) using the contrast image [NC > (EC, PC)] (based on differential attribute representation for $Self for this comparison). Resulting pattern vectors (one per subject) were used as input features for the prediction, and individual difference scores in regulatory success served as labels. Regulation success was defined using difference scores in observed choice behavior (e.g., ΔGenerous choices [NC - (EC, PC)]) and in DDM parameters (e.g., Δw Self [NC - (EC, PC)]). Predictions used a linear ν-SVR (libSVM) with a fixed cost parameter c = 0.01 (similar to within-subject decoding) and a leave-one-subject out approach (yielding a 36-fold cross-validation). Decoding accuracies reflect correlations of the observed and predicted scores of individuals' regulatory success. Statistical significance was assessed by comparisons of decoding accuracies to empirical null-distributions (realized by randomly permuting the pairing of participants' neural pattern vectors and behavioral regulation scores 1000 times). Only

decoding accuracies above the 95th percentile of null-distributions were considered statistically significant. See Supplemental Table 5 for results.

## DLPFC-based prediction of goal-consistent changes of generosity is driven by goal-consistent changes in attribute representations of $Self (but not $Other or Fairness)

On the one hand, neural activation patterns in the DLPFC flexibly encoded values of tastiness, healthiness, and $Self, but not $Other or $Fairness. On the other hand, the DLPFC strongly predicted goal-consistent change in altruistic choices. Is the DLPFC prediction of altruistic choices then mediated especially by the change in $Self during altruistic choices, but not by the change to $Other or $Fairness?

We addressed this idea in the following way. First, at the behavioral level, we regressed changes in generosity (ΔGenerous Choices [(PC, EC) - NC]) on changes in model-based weights for subject's own benefits (Δw $Self [NC - (EC, PC)]), and calculated the residuals. This gave us a measure of change in generosity after controlling for change in weight on $Self. Next, we repeated the cross-subject decoding analysis (SVR) using neural response patterns in the DLPFC as features and the estimated residuals as labels for the prediction. This tells us whether the DLPFC continues to predict altered generosity after controlling for change in the input of $Self on choices. Our results are consistent with the idea that the DLPFC predicts change in generosity because it encodes changes in weight on $Self: activation patterns in the DLPFC no longer predicted individuals' goal-consistent changes in generosity when we controlled for changes in the weight of self-related benefits on choices ($r = 0.11$, $p = 0.317$, permutation test, [CI: −0.41, 0.41]). We next sought to show that this effect is *specific* to changes in self-related considerations. To do this, we repeated this supplemental analysis for each of the other choice attributes (i.e., regressing out change in weight on $Other or change in weight on Fairness, and then asking whether DLPFC predicts residual changes in generosity). We found that DLPFC prediction of altered generosity remained highly significant when we controlled for altered inputs of other's benefits ($r = 0.51$, $p = 0.016$, [CI −0.41, 0.40]) or fairness considerations ($r = 0.56$, $p = 0.007$, [CI −0.39, 0.39]) on choices. Taken together, these findings indicate that predictions of goal-consistent changes in generosity in the DLPFC are mediated specifically by altered weighs of subjects' own monetary benefits on choices (Δw $Self). This finding is also consistent with evidence of a specific functional role of the DLPFC for altered representations for self-related considerations, while changes in attributes of $Other and Fairness are encoded elsewhere.

## Changes in functional connectivity with the VMPFC correlate with regulatory success

Results reported in the main text pose an important question: if attribute encoding within certain brain regions changes as a function of regulatory goals, how are such changes accomplished? We speculated one of two possibilities. First, we observed changes in attribute encoding not only within the DLPFC and social-cognitive brain areas (e.g. Precuneus, TPJ), but also within areas of visual cortex for some contrasts. This suggests that altered functional coupling between goal-sensitive areas and visual cortex could be producing some of the changes observed in attribute encoding. Alternatively, given that the VMPFC encoded all choice-relevant information, but showed no modulation as a function of goal, we speculated that the VMPFC might be differentially connected to these goal-sensitive areas as a function of regulatory success. To test these different hypotheses, we examined regulation-associated changes in connectivity patterns of the DLPFC conjunction area, as well as the TPJ and Precuneus, focusing specifically on evidence for changes in connectivity within functionally-defined masks of the occipital or motor cortices and the VMPFC. We also performed the reverse analysis, examining changes in functional connectivity of the VMPFC with other areas.

We used a beta series approach to functional connectivity (*Rissman et al., 2004*). In brief, this method uses a GLM to estimate the BOLD response in a region, separately for each trial in a condition of interest, and after controlling for other variables of non-interest (e.g., response on other trial, response in other conditions, motion, etc.). The resulting beta coefficients represent the estimated neural response in that region on each trial. These values are then included as a parametric modulator in a new GLM, to identify brain regions where response systematically covaries with the trial-by-trial fluctuations in the ROI. We can extract these 'beta-series' for each condition of interest (e.g., Health Focus, Taste Focus, Natural Response) and then compare whether neural correlates of trial-by-trial fluctuations in an ROI's response vary as a function of condition. We used this approach rather than more traditional psychophysiological interaction (PPI) analyses, because recent work suggests that this method may be more sensitive to contextual differences in functional connectivity than more traditional PPI approaches (*Rissman et al., 2004*). We provide greater details of this analysis below.

## Methods: Exploratory functional connectivity analyses

### Beta series estimation

The first step in a beta series analysis is to extract estimates of trial-by-trial response in an ROI for conditions of interest, controlling for other experimental factors.

Thus, we estimated trial-by-trial responses in each regulation condition for four different regions-of-interest: 1) the VMPFC area demonstrating a conjunction in coding of all attributes regardless of regulatory goal; 2) the DLPFC area showing a conjunction in regulation effects for Tastiness, Healthiness, and $Self attributes; 3) and 4) the TPJ and Precuneus ROIs showing changes in response to $Other in Partner vs. Ethics trials. To do this, we conducted a GLM consisting of the following regressors: R1-R90: Indicator functions delineating the choice period on each separate trial for one of the regulation conditions (e.g. Natural trials); R91 and R92) Indicator functions indicating the choice period for the two non-target regulation conditions (e.g., Focus on Health, Focus on Taste); R93-R96) Parametric modulators of R91-92 consisting of health and taste ratings for the food shown on each trial; R97) An indicator function for missed response trials. This model allowed us to extract trial-specific estimates of neural response in an ROI for a specific regulatory condition, controlling for activation related to other trials and conditions. These estimates were created by averaging the beta-coefficient for each trial (R1-R90) across all voxels for each of the four target ROIs listed above, which served as our signal of interest. This process was repeated once each for the other two regulation conditions, yielding 270 beta estimates, one for each trial in each condition.

A similar beta series analysis was conducted for the Altruism Task, with the exception that the model also included regressors for the outcome period on each trial, separately for each regulation condition (R97-99), as well as parametric modulators of R97-99 representing the monetary outcome for Self (R100-102) and Other (R103-105).

We also extracted trial-by-trial responses from a mask of the whole brain, which allowed us to control for non-specific global hemodynamic changes in response, and to focus on connectivity specific to a given region of interest.

### Beta series connectivity analysis

The next step of a beta series analysis involves using the estimated trial-by-trial betas in a new GLM to identify areas where response correlates with trial-by-trial fluctuation in a given ROI. Details of the GLMs we ran for each ROI are as follows:

### GLM S3: Changes in connectivity of the DLPFC during cognitive regulation of food choices

To examine connectivity of the DLPFC, the extracted trial-by-trial beta estimates were entered into a model identical to GLM S1 (described in detail in Supplemental Material 3.7 below), but including six additional parametric modulators. (R10-12) consisted of the trial-by-trial estimates of whole-brain response, and served as a regressor of non-interest that controlled for non-

specific hemodynamic changes. (R13-15) consisted of the critical variables of interest: the trial-by-trial beta estimates from the DLPFC ROI during NC trials (R13), TC trials (R14) and HC trials (R15). This allowed us to examine changes in functional connectivity with the DLPFC ROI after controlling for other features of the experimental design, including systematic variation in attribute values.

Contrast estimates were calculated at the individual subject level for the following contrasts: R14 – R13 (DLPFC connectivity in Taste vs. Natural trials), R15 – R13 (DLPFC connectivity in HC vs. NC trials), and 2*R15 – [R13 +R14] (DLPFC connectivity in HC vs. NC +TC trials). Group-level t-tests were performed on these contrast estimates to determine overall differences in connectivity by condition. In addition, we also estimated the correlation between these contrasts and the corresponding change in behavior weights (e.g. Δ Healthiness weight, HC > [NC +TC]), to determine whether changes in functional connectivity predicted regulatory success for attributes in the food choice task. Since this was an exploratory and supplemental analysis, we report results at a significance level of p < 0.005, uncorrected for voxels falling within ROIs of interest (e.g. the VMPFC conjunction area).

## GLM S4: Changes in connectivity of the VMPFC during cognitive regulation of food choices

This GLM was identical to GLM S3, with the exception that it used trial-by-trial responses extracted from the VMPFC during the food choice task.

## GLM S5: Changes in connectivity of the TPJ during cognitive regulation of altruistic choices

This GLM was identical to GLM S2, except that it included the trial-by-trial beta estimates from the TPJ ROI during the NC condition (R13), PC condition (R14) and EC condition (R15). All other details are similar to GLM S3.

Contrast estimates were calculated at the individual subject level for the following contrasts: R14 – R13 (TPJ connectivity in PC vs. NC), R15 – R13 (TPJ connectivity in EC vs. NC), R14 – R15 (TPJ connectivity in PC vs. EC trials), and [R14 + R15] – 2*R13 (TPJ connectivity in both social focus condition vs. NC trials). Group-level t-tests were performed on these contrast estimates to determine overall differences in connectivity by condition. In addition, we also estimated the correlation between these contrasts and the corresponding change in behavior weights (e.g. ΔOther weight, PC > EC), to determine whether changes in functional connectivity predicted regulatory success for attributes in the altruistic choice task. Since this was an exploratory and supplemental analysis, we report results at a significance level of p < 0.005, uncorrected for voxels falling within ROIs of interest (e.g. the VMPFC conjunction area).

## GLM S6: Changes in connectivity of the Precuneus during cognitive regulation of altruistic choice

This GLM was identical to GLM S5, with the exception that it used trial-by-trial responses extracted from the Precuneus during the altruistic choice task.

## GLM S7: Changes in connectivity of the VMPFC during cognitive regulation of altruistic choice

This GLM was identical to GLM S5, with the exception that it used trial-by-trial responses extracted from the VMPFC during the altruistic choice task.

## Results: Exploratory functional connectivity analyses

### DLPFC connectivity during food choices

We began by examining connectivity patterns during the food choice task using the DLPFC region described by the conjunction of goal-consistent changes in Tastiness, Healthiness, and $Self attributes. We expected that altered connectivity of the DLPFC with VMPFC or visual areas during the HC focus condition, and that changes in connectivity with these areas should

correlate with increases in weighting of the Healthiness attribute during the Health Focus condition, decreases in weighting of the Tastiness attribute, or both.

We did not observe a main effect of cognitive regulation of decision making on functional connectivity to either visual/motor areas or VMPFC, even at a liberal statistical threshold of p < 0.005 (uncorrected). However, we found that functional connectivity with the VMPFC correlated with regulatory success. In particular, successful up-regulation of behavioral health weights (Δw Healthiness, (HC - [NC, TC])) correlated with increased connectivity to VMPFC (p < 0.005, uncorrected, *Figure 3—figure supplement 2A*) when subjects focused on Health in the food task compared to natural or taste trials. No correlations emerged between connectivity to the visual cortex and regulatory success.

### DLPFC connectivity during altruistic choices

Given that we observed decreased encoding of selfish outcomes in the DLPFC during both PC and EC compared to NC trials, we expected either that we should observe a decrease in connectivity between DLPFC and VMPFC or visual areas during these conditions, or that decreased connectivity should correlate with decreases in the weighting of Self during regulation trials. As in the Food Task, we observed no significant alterations in connectivity overall. However, as predicted, decreased weighting of $Self (Δw $Self, (NC – [EC, PC])) correlated with decreased coupling of the DLPFC with VMPFC (p<0.005, uncorrected, *Figure 3—figure supplement 2B*) during regulation trials. No correlations emerged between connectivity to the visual cortex and regulatory success.

### TPJ and Precuneus connectivity during altruistic choice

We observed specific changes in coding of others' outcomes in the TPJ and Precuneus when focusing on one's partner's feelings (PC) compared to focusing on ethics (EC). We thus expected that we should observe changes in coupling between TPJ or Precuneus and other areas, either overall, or as a function of changes in weighting of partner's outcomes. We observed no overall effects on connectivity in either region. In contrast, tighter coupling between the Precuneus and VMPFC in Partner vs. Ethics trials correlated with greater weight given to $Other in Partner vs. Ethics trials (p <0.005, uncorrected, *Figure 3—figure supplement 2C*). No correlations emerged between connectivity to the visual cortex and regulatory success.

## Univariate Analysis of fMRI Data

*GLM S1.* We compared the multivariate SVR results (Support Vector Regression) with those of a traditional univariate GLM based on data smoothed at 8 mm FWHM. For each subject in the food task, we estimated GLM S1 with AR(1) and the following regressors of interest: R1) A boxcar function for the choice period on Natural trials; R2) R1 modulated by the taste rating for the food on each trial; R3) R1 modulated by the health rating for the food on each trial; R4-R6) analogous regressors for HC trials; R7-9) analogous regressors for TC trials. No orthogonalization was used, allowing regressors to compete fully for explained variance. All regressors of interest were convolved with the canonical form of the hemodynamic response. The model also included motion parameters and session constants as regressors of no interest. Missed response trials were modeled as a separate regressor of no interest.

Subject-level contrast images were calculated for each of the parametric regressors (R2, R3, R5, R6, R8, and R9) and used for separate random-effects group analyses. We used one-sample t-tests as implemented in SPM to look for areas where linear variation in BOLD response correlated with trial-to-trial variability in attribute values, and paired t-tests to test for modulation of BOLD responses by condition. Repeated measures ANOVAs were used to examine if neural responses encoding choice-attributes varied across regulation conditions (p < 0.05, FWE cluster-corrected, p < 0.001 height threshold).

*GLM S2.* We computed a second, analogous GLM to estimate univariate responses in the altruism task. This GLM was identical to GLM S1 with the following exceptions: 1) Natural, Taste, and Health conditions were replaced with Natural, Ethics, and Partner Focus conditions; 2) parametric modulators for taste and health were replaced with the value of $Self and

$Other on each trial; 3) we included a third parametric modulator in each condition to represent fairness $-1*|$Self - $Other|$ on each trial. All other details are as in GLM S1.

See *Supplementary file 1E* for results for univariate analyses.

## Self-reported motivation to comply with instructions and observed regulation-success

One potential concern with the findings described in the main text is that they result purely from experimental demand effects, and have no bearing on real-world behavior. Although such demand effects were likely reduced by the probabilistic implementation of participants' choices (in effect obscuring their choice from the experimenter), we examined if individual differences in regulatory success in either choice task merely reflect peoples' tendency to comply with demands of the current experimental condition by examining self-reported compliance for the sake of the experimenter.

We obtained self-report measures of experimental demand by asking participants 'How much did you try to choose based on the experimenters expectations?' (5-point scale, 1='not at all', 5='absolutely'). These indicated that experimenter demand effects were generally low in our sample of participants (M ± SD: 1.83 ± 1.21, range of 1 to 4). Moreover, we found no significant correlation with regulation success in the food task (all p's > 0.14, uncorrected) or the altruism task (all p's > 0.16, uncorrected). This suggests that individual differences in observed healthy and generous choices were not explained purely by demand characteristics of the tasks.

