## [Decision Letter]

Thank you for submitting your article "Computations for altered attribute representations underlie cognitive regulation in altruistic and healthy choices" for consideration by *eLife*. Your article has been favorably evaluated by Michael Frank (Senior Editor) and three reviewers, one of whom is a member of our Board of Reviewing Editors. The reviewers have opted to remain anonymous.

The reviewers have discussed the reviews with one another and the Reviewing Editor has drafted this decision to help you prepare a revised submission.

Summary:

In this study, the authors explore the ability of goals or attentional focus to modulate choices in a food task and in an altruism task. In the tasks, subjects chose or rejected different foods or different monetary offers. Instructions to choose healthy or to make generous choices modulated the behavior in both tasks and also affected neural activity patterns measured using fMRI. Regulatory success – or the ability to use instructions to change default choice behavior – was correlated with DLPFC activity.

Essential revisions:

One reviewer, not someone doing fMRI work, had significant difficulty simply following how the concepts of value versus regulation were operationalized in the task and analysis. What precisely were the conditions compared to isolate regulation for example from value? Describing this in more concrete terms would be helpful. This was related to the overarching concern of the other reviewers as well, which was the distinction between the authors conceptualization versus a more trivial interpretation – namely that when participants pay attention to a particular type of information (by instruction), this information is easier to decode from their brain activity. As noted below: "This effect should be seen as the outcome (and not the process) of cognitive regulation. How cognitive regulation is implemented, and how better decoding translates into biased choices, still need to be explained." Basically what is needed is to provide a more mechanistic account for what is going on.

Suggestions are given in reviews below. For example "it would be important to show that the decoded value (i.e., the decoder output) correlates with the behavioral weights. The alternative would be that changes in decoding accuracy correspond to changes in precision (i.e., signal-to-noise ratio) and not changes in the signal itself. If correct, this would mean that a region downstream to the DLPFC could just read this value, add it to other values corresponding to other attributes, and feed the aggregate value to a selection process that makes the decision. Perhaps functional connectivity could be used to test for such a transfer of information. Thus, the neural model would parallel the behavioral model."

Another suggestion was that some sort of functional connectivity analysis could clarify *how* DLPFC (and potentially rTPJ) enables successful regulation mechanistically, possibly using additional ROI analyses on VMPFC to ensure this is indeed the case.

These are the essentially revisions or problems identified in our discussion – clarify the approach and provide a more mechanistic account of the proposed interaction to rule out trivial explanations of the findings.

*Reviewer #1:*

I think the general question is of interest and the authors approach is quite novel to me. They are basically testing how changes in goals affect behavior. This is similar to the use of devaluation in animal learning theory tasks, but here they are using simple instructions in the course of training with humans. I think this is a creative way to integrate economic decision making, which focuses on a unitary utility as guiding choices, with work from experimental psychology and computational neuroscience, which typically distinguishes different sources of information by its associative basis or computational basis. This is excellent.

However, beyond that I had difficulty following the authors framing, predictions and understanding the outcomes of the neural activity analysis. The DLPFC is the key "regulatory" area. But what does that mean? It seems to me that I would expect some areas to represent value independent of goals and some areas to represent that value only when goal relevant. Is this what is meant by regulatory? Does the DLPFC represents value relevant to goals and some other area does not – VMPFC?

Generally I think the question is very interesting and the approach is attractive, but I simply could not follow how the authors framed and then conducted their analysis. I will be interested to see what the other reviewers say. I might grasp things better if it were more clear how the theoretical concepts were operationalized for the analyses – precisely what is "regulatory" for example and how is it distinguished from non-regulatory versus just not involved by the task and then in the data analysis.

*Reviewer #2:*

Tusche and Hutcherson present a thought-provoking and methodologically impressive study on cognitive regulation of dietary and altruistic choices, a topic of broad interest. The analyses, which combine a drift diffusion computational model of attribute-weighted choice and trial-by-trial MVPA decoding, are sophisticated, appropriate, and comprehensive. They show that while attribute values across choice and regulatory goal contexts can be decoded in VMPFC, they do not appear to be modulated by regulatory goal. Conversely, attribute values in DLPFC are modulated consistently with model-derived behavioral weights across regulatory goals that emphasize either healthiness or tastiness during dietary choice, and personal gain in altruistic choice contexts. Regulatory goals that emphasize another's benefits (e.g. feelings), however, could instead be decoded from right TPJ and precuneus, but not DLPFC, suggesting representations of others' wellbeing is modulated according to the prosocial regulatory goal but only when it requires theory of mind. These findings speak to both the domain-generality and domain-selectivity of cognitive regulation of decision making and importantly advance our understanding of the neural systems important for cognitive control and decision making more generally. In general I am supportive of this paper but think some outstanding issues could be better addressed to confirm some of their interpretations and rule out others.

It is notable that DLPFC flexibly encoded values of tastiness, healthiness, and $Self but not $Other or $Fairness on the one hand but strongly predicted altruistic choices on the other hand. Is the DLPFC prediction of altruistic choices then mediated especially by the change in $Self during altruistic choices, but not the change to $Other or $Fairness?

A related question: Do the rTPJ and precuneus group effects predict individual differences in regulatory success during altruistic choices when goals depended on another's thoughts or feelings? Such evidence would tie together their argument that these latter regions "assume responsibility" for regulatory success when DLPFC does not because of the component process required to meet that goal.

Given the prior literature on this topic, and in order to rule out a model whereby VMPFC value representations are modulated in an attribute-specific manner that depends on regulatory goals, it would be informative to see an ROI-based analysis of VMPFC. I may have missed something but I could not find one. The ROI from the conjunction presented in Figure 3 could be used or prior literature could be used.

One question which is unclear from the conjunction analyses is whether it is the same neural code (e.g. in DLPFC) that is used across task contexts, or whether the code is distinct (context-dependent) but found in the same brain region. What do the authors find if the SVR is trained on one attribute (e.g. tastiness) and tested on the other two (e.g. healthiness and $self)? Can the overlap between representations be better visualized?

It would be informative to visualize the feature weights for the key areas (e.g. VMPFC, DLPFC) across voxel space in every subject. This procedure should help to assess to what extent any decoding effects are due to hard anatomical boundaries between subareas (e.g. dorsal and ventral aspects of DLPFC) or to distributed patterns within areas.

How do the authors interpret the altered representations in some primary sensory and motor areas, e.g. less strong decoding of tastiness values for NC than HC and TC? Have the authors considered a model whereby coupling between DLPFC and distinct regions of sensory cortex is modulated according to the regulatory goal? The DLPFC must get its attribute representations from someplace.

The one comparison in which a match with the behavioral analyses failed to emerge was for ethical considerations compared to normal or personal considerations. I do not view this null finding as problematic, but out of curiosity do the authors have any thoughts as to what is going on when regulatory success depended on changes to fairness due to the goal of complying with social norms? This null finding should be addressed in the Discussion.

*Reviewer #3:*

In this manuscript, Tusche and Hutcherson report an fMRI study on how cognitive regulation affects the neural representation of choice-relevant attributes. They look for mechanisms that may generalize across two types of choices, one involving conflict between healthiness and tastiness of food items, the other involving conflict between self-interests and altruistic concerns. In different conditions, participants are asked to focus on one or the other attribute, which regulates the weights assigned to the targeted attributes, as shown via computational modeling of choice data. The key findings are the links between these changes in attribute weights and the decoding accuracy obtained for these attributes using multivariate pattern analysis (MVPA) in various cortical regions. The results are quite convincing, with successful decoding across tasks and individuals. There is no clear conclusion about whether the regulation is centralized or distributed though, since changes in decoding accuracy are observed in the DLPFC for most of the attributes but not all.

The role of cognitive control in economic choice is poorly understood and this study brings valuable insights by applying MVPA to standard choice paradigms. My main concern is the absence of a mechanistic account linking brain activity to behavioral output. In a sense, the results seem a bit trivial: when participants pay attention to a particular type of information (by instruction), this information is easier to decode from their brain activity. This effect should be seen as the outcome (and not the process) of cognitive regulation. How cognitive regulation is implemented, and how better decoding translates into biased choices, still need to be explained.

For the latter point, it would be important to show that the decoded value (i.e., the decoder output) correlates with the behavioral weights. The alternative would be that changes in decoding accuracy correspond to changes in precision (i.e., signal-to-noise ratio) and not changes in the signal itself. If correct, this would mean that a region downstream to the DLPFC could just read this value, add it to other values corresponding to other attributes, and feed the aggregate value to a selection process that makes the decision. Perhaps functional connectivity could be used to test for such a transfer of information. Thus, the neural model would parallel the behavioral model.

Other points:

- The correlation across individuals could reflect compliance to the instructions rather than self-regulation capacity. The arguments taken from subjective report and from body-mass index are quite weak. For subjective report it could be that the rating scale is not reflecting the propensity to comply with the instructions. For body-mass index the opposite correlation could be expected: those who regulates food intake in real life should not need instructions in the lab.

- The observation that all attributes are represented in the VMPFC but inaccessible to cognitive regulation is super interesting (and novel, to my knowledge). The dissociation with DLPFC should be more emphasized and discussed. Would this mean that VMPFC representations are closer to stimuli and DLPFC to responses?

- To compare the pattern of attribute weights and the pattern of decoding accuracy across conditions, the authors intend to reproduce significance of pair-wise comparisons. As they know this approach heavily depends on the statistical threshold, which may be matter of debate when multiple comparisons are made. I would favor a straight regression of decoding accuracy against weight (across conditions).

---

## [Author Response]

Essential revisions:One reviewer, not someone doing fMRI work, had significant difficulty simply following how the concepts of value versus regulation were operationalized in the task and analysis. What precisely were the conditions compared to isolate regulation for example from value? Describing this in more concrete terms would be helpful.

We thank the editor for the opportunity to resubmit our paper, and for the thoughtful comments and suggestions. We agree with reviewer 1 that we were not as clear as we could have been in the previous instantiation of the manuscript about the focus of our paper, and on the distinctions between value and regulation. We have substantially revised the Introduction (as described in greater detail below, see our first response to reviewer 1) to make this clearer. In brief, we have more clearly specified that we are focused on disentangling different hypotheses about what stage of the processing stream regulation acts on to modify value computations and have explicitly spelled out different possibilities along with the predictions they make at the neural level. Moreover, we have substantially revised the Results section of the manuscript to orient the reader to how different analyses address these different questions. Finally, we added a figure to illustrate the decoding analysis that aimed to identify brain regions that encode value in a goal-dependent manner (Figure 4) as well as a panel in Figure 1 to illustrate the behavioral computational modeling approach (Figure 1D). We feel that these revisions have significantly strengthened the clarity of the manuscript.

This was related to the overarching concern of the other reviewers as well, which was the distinction between the authors conceptualization versus a more trivial interpretation – namely that when participants pay attention to a particular type of information (by instruction), this information is easier to decode from their brain activity. As noted below: "This effect should be seen as the outcome (and not the process) of cognitive regulation. How cognitive regulation is implemented, and how better decoding translates into biased choices, still need to be explained. " Basically what is needed is to provide a more mechanistic account for what is going on.

Regarding the first point, we agree that in some sense it should be “trivial” to observe changes in information decoded from brain activity that matches changes at the behavioral level. Nevertheless, as we make clear in the revised Introduction and Discussion sections, extant research has actually not consistently found such patterns despite clear changes in behavior, raising an important puzzle to be explained. Our results are the first to identify a clear representation of all choice-relevant attributes across multiple contexts, and to show clear modulation of these choice-relevant attributes in a manner consistent with regulatory goals.

We also think our specific pattern of results clearly speak against the trivial interpretation that one would have to observe changes in attribute coding in the brain, and that this is an uninteresting reflection of changes in behavior that gives no insight into mechanism. We observe a highly specific pattern of altered and unaltered representations of attribute values, which reveals important dissociations across different kinds of attributes and different individuals. This is anything but trivial, and, we believe opens up crucially important lines of questioning that will be the basis for a better understanding of cognitive regulation and its mechanistic implementation. We have substantially revised the Introduction and Discussion to highlight these important points (see our specific responses to reviewers and our next response below).

Regarding the second point (that this should be seen as the outcome and not the process, and that this does not reveal mechanism), we respectfully disagree that no information about mechanism can be derived from a detailed description of where such changes are observed. We have tried to make this clearer in the following ways:

1) We have now further clarified in the new version of the paper in the Introduction, Results, and Discussion how identifying where attribute encoding changes occur may allow us to distinguish between different theories about computational process and mechanisms. Specifically, we make clearer that regulation could occur via two mechanisms: modulation at the level of specific attribute representations in non-overlapping, attribute-specific regions, or at the level of weighting in the value integration process.

2) We also make clearer that there are important outstanding questions about the domain-generality or domain-specificity of regulation effects. On the one hand, if different attributes are computed in different regions with different computational properties, regulation could operate through different channels in different domains. If, instead, regulation operates at a more domain-general level, then regulation might influence attribute representations in a common region across multiple different attributes and multiple choice contexts. Understanding whether regulation operates in a domain-general or domain-specific way can help reveal both the targets of regulation, as well as the mechanism of action.

3) In the Discussion, we highlight how these questions are not of purely theoretical importance: the locus of action (attribute level or integration level, domain specific or domain general) predicts non-trivial differences in patterns of regulatory success, which we observe in our own data. For attributes where regulation influences representation in a common region (i.e., DLPFC) regulatory success tracks together, but for attributes represented elsewhere (i.e., $Other in TPJ and precuneus), regulatory success appears to operate independently. Our results also may help to explain when and why regulatory success might feel difficult: if, as we observe, there are many regions of the brain where attribute representations are not altered by regulation, then these altered representations may continue to leak into the choice process and interfere with regulatory success.

4) We have also included several supplementary analyses, including functional connectivity results, which shed some light on mechanistic processes.

For these reasons, we believe this paper will be of considerable interest to the readership of *eLife*, and have tried to articulate this reasoning better in the paper. We hope the reviewers and editor will agree.

Suggestions are given in reviews below. For example "it would be important to show that the decoded value (i.e., the decoder output) correlates with the behavioral weights. The alternative would be that changes in decoding accuracy correspond to changes in precision (i.e., signal-to-noise ratio) and not changes in the signal itself. If correct, this would mean that a region downstream to the DLPFC could just read this value, add it to other values corresponding to other attributes, and feed the aggregate value to a selection process that makes the decision. Perhaps functional connectivity could be used to test for such a transfer of information. Thus, the neural model would parallel the behavioral model."Another suggestion was that some sort of functional connectivity analysis could clarify how DLPFC (and potentially rTPJ) enables successful regulation mechanistically, possibly using additional ROI analyses on VMPFC to ensure this is indeed the case.

We agree with the editor and reviewer that each of these issues is incredibly important, and have sought to address them in a series of supplementary analyses.

1) First, we used regression of decoding accuracies against behavioral weights (as suggested by reviewer 3). We show that results corroborate findings from the ANOVAs and paint essentially the same picture in our regions of interest. Specifically, we find goal-consistent coding of attributes that match predictions of the behavioral model. We therefore believe that these results are not simply a reflection of changes in signal-to-noise. However, we also believe that the ANOVAs actually give us greater power to detect effects, particularly in the altruistic choice task, where it seems clear that our behavioral weights (particularly for a construct like other’s welfare) reflect different considerations in the different regulatory focus conditions. These ANOVAs reveal, for instance, that TPJ and Precuneus seem to code for others’ welfare when focused on partner feelings, but not when focused on ethical reasons for giving, despite increased behavioral weighting on $Other in both. This is an important result in its own right, and sheds light on the computational function of these regions. This result would not have been clear using the regression analysis proposed by the reviewer. We have thus opted to retain the original analyses, but provide a detailed description of the regression analyses in our point-by-point response, to corroborate and extend the conclusions that we come to using the ANOVAs. We opted for this course to maintain a clear and streamlined set of results, and to avoid over-taxing the reader’s working memory, since the paper is already rich in terms of methodological approaches and results. If the editor or reviewers deem it appropriate and necessary, we can of course include the regression analyses in the main body of the paper.

2) Second, we have now performed a detailed set of functional connectivity analyses to test for different theories about the role of information transfer in producing the observed pattern of results. Although we believe that functional connectivity analyses are somewhat complicated to interpret in the context of multivariate rather than univariate analysis (see our more detailed response to reviewer 3), we have also taken to heart the reviewers’ suggestions to dig deeper into our data. Thus, we have now conducted exploratory functional connectivity analyses to test whether and how changes in patterns of functional connectivity between regions are linked to regulatory success. To do this, we used a beta series approach (Rissman et al., 2004), which research suggests may be more sensitive to modulation of functional connectivity by context in studies like ours that use an event-related design with many trial repetitions (Cisler et al., 2014).

In brief, we extracted trial-by-trial fluctuations of BOLD response in regions of interest (i.e., DLPFC, precuneus, TPJ, and VMPFC) and then looked for areas of the brain where correlation with this trial-by-trial variation (after controlling for whole-brain response) varied as a function of regulatory goal. We looked both for main effects at the group level, as well as effects that correlated with regulatory success (e.g. identifying functional connectivity differences in HC vs. NC that correlated with change in behavioral weights of the healthiness attribute in HC vs. NC). At the suggestion of reviewer 2, we also specifically focused on two possibilities in terms of altered functional connectivity. First we tested the hypothesis that DLPFC, TPJ, and/or precuneus show altered connectivity with visual and/or or motor areas, and that this explains observed differences in regulation success. Second, we tested the hypothesis that DLPFC, TPJ and/or precuneus show altered connectivity with the VMPFC.

Our results most clearly support the latter hypothesis. At admittedly-somewhat-liberal thresholds, we find hints that connectivity patterns between VMPFC and DLPFC as well as between VMPFC and Precuneus correlate with changes in behavioral weighting of choice attributes in a manner that is consistent with predictions of our behavioral computational model. Using the DLPFC as a seed region, we find that increased connectivity with VMPFC correlates with greater weighting of the healthiness attribute when focused on health during food choice, and lower weighting of the self outcomes when focused on pro-social motives in the altruism task. Using the precuneus as a seed region, we find that increased connectivity with the VMPFC correlates with greater weighting of partner outcomes in the contrast of Partner vs. Ethics focus (which is also the contrast that most clearly differentiates encoded neural information on partner outcomes ($Other) in precuneus). A similar analysis using the VMPFC as a seed region revealed similar results.

What might these results mean in terms of mechanism? Despite the functional coupling with the VMPFC, representations of these attributes values within the VMPFC did not change (as confirmed by three supplemental analyses, see our fourth response to reviewer 2). This finding suggests that altered functional connectivity did not result in altered goal-consistent attribute representations in the VMPFC. Instead, our results raise an intriguing (albeit speculative) possibility as an alternative mechanism: signals from the VMPFC may actually be used to alter computations within DLPFC and Precuneus to amplify or diminish attribute values computed in these areas. Although this is not the modal view of the function of the VMPFC, it is actually consistent with some early work on emotional conflict resolution suggesting that the VMPFC may be involved in modulating affective representations in lower-level areas such as the amygdala in a goal-consistent way (Etkin et al., 2006; Quirk and Beer, 2006).

Because these results were observed only at relatively liberal thresholds, we have opted not to make them a primary focus of the main paper. However, we have included them in the Appendix and refer the interested reader to them in the appropriate portions of the manuscript. We hope that this approach balances our twin goals of keeping the main paper streamlined and informative, while also providing full disclosure about analyses that may be of interest.

These are the essentially revisions or problems identified in our discussion – clarify the approach and provide a more mechanistic account of the proposed interaction to rule out trivial explanations of the findings.

As outlined above, we have thoroughly revised the manuscript to further clarify our approach. We have also added several new analyses to provide a more mechanistic account of the proposed interaction of identified brain regions, and to rule out other explanations of the findings. We believe that these revisions have substantially improved the manuscript and we truly appreciate the editors’ and reviewers’ constructive and thoughtful suggestions. Please find below the point-by-point reply to each comment of the reviewers.

Reviewer #1:I think the general question is of interest and the authors approach is quite novel to me. They are basically testing how changes in goals affect behavior. This is similar to the use of devaluation in animal learning theory tasks, but here they are using simple instructions in the course of training with humans. I think this is a creative way to integrate economic decision making, which focuses on a unitary utility as guiding choices, with work from experimental psychology and computational neuroscience, which typically distinguishes different sources of information by its associative basis or computational basis. This is excellent.However, beyond that I had difficulty following the authors framing, predictions and understanding the outcomes of the neural activity analysis. The DLPFC is the key "regulatory" area. But what does that mean? It seems to me that I would expect some areas to represent value independent of goals and some areas to represent that value only when goal relevant. Is this what is meant by regulatory? Does the DLPFC represents value relevant to goals and some other area does not – VMPFC?

We thank the reviewer for his/her positive comments and enthusiasm for the approach! We apologize for the lack of clarity in the previous version of the manuscript and appreciate that the reviewer brought this to our attention. In order to fully address the comments, we have broken down our responses into three separate but related issues: framing, predictions, and understanding of outcomes.

A) Framing.

Our usage of the term “regulatory” in the original version of our manuscript needed further clarification. In any study of cognitive regulation (e.g. in the context of research on emotion, or decision making, or perceptual tasks), it is important to distinguish between an effector and an effect of regulation. The former refers to the causal driver of changes, while the latter is in essence the result of regulatory efforts. Revealing the neural effectors and effects of regulation are related but separate goals in neuroeconomics. Our analyses were designed specifically to focus on the latter: how do value representations (either at the level of individual choice-relevant attributes or at the level of integrated decision values) change as a function of regulatory goals?

We have substantially revised and reframed the introduction to the paper to clarify this point and to highlight what we might learn by understanding how regulation changes value representation. More specifically, we now spell out in greater detail two different hypotheses about the effects of regulation:

The attribute-level hypothesis suggests that regulatory goals operate by altering computations in brain areas specialized for processing specific attributes. For example in the food task, there might be specific regions that represent the current value of a food’s tastiness and other regions that represent the value of its healthiness. If regulation operates at this attribute level, then each of these specific regions would be independently modulated depending on goals. This view suggests that we should see alterations in attribute-specific areas. On the other hand, the integration-level hypothesis suggests that regulatory goals operate by altering weighting of those attributes only in an area (or areas) representing decision values. If regulation operates only at the integration-level, then goal-consistent alterations in attribute representation should be observed only within centralized, domain-general regions associated with value integration and decision value computations (i.e., VMPFC). This hypothesis thus predicts that attribute-specific areas should represent attributes regardless of regulation, while value integration regions might shift to flexibly represent attribute in a goal-consistent manner.

We have now worked to re-frame the Introduction, focusing on this distinction, as well as two corollary questions: 1) Does the level at which regulation operates (i.e., attribute-or integration-level) depend on the specific attribute and/or domain? 2) What role do the VMPFC and DLPFC play in these processes? We believe that these revisions have substantially improved the clarity and focus of the manuscript, as well as highlighting its major contribution to the literature on regulation.

The relevant section of the revised manuscript now reads:

“A key goal of neuroeconomics is to describe how these attribute and decision value computations change as a function of regulatory goals and contexts, and to link such changes to regulatory success. Here, we sought to address three important questions about this process […] However, several failures to observe changes in the VMPFC during cognitive regulation of decision making (Hollmann et al., 2012; Hutcherson et al., 2012; Yokum and Stice, 2013) suggest the need to either measure value computation in a more sensitive way, or to identify alternate routes to behavioral change.”

We have also revised the title of the manuscript and the Abstract to further clarify that the study is interested in the effect of cognitive regulation on value representations.

The revised title is:

“Cognitive regulation alters social and dietary choice by changing attribute representations in domain-general and domain-specific brain circuits”

B) Predictions.

We have substantially revised the manuscript to better link the analyses and our hypotheses. Both the attribute-level and integration-level theories predict that goal-dependent value signals will be represented more strongly when participants’ regulatory goals emphasize those attributes. The critical question is where this modulation is observed, as this allows us to distinguish between the two hypotheses outlined above. We have added three overview paragraphs to the revised Results section that emphasize the overall aim of the subsequent set of analyses and the predictions more clearly.

“Behavior

To identify how value computations change to accommodate regulatory goals, our analysis strategy proceeded in the several steps. […] We also used our computational behavioral models (multi-attribute drift diffusion models, DDMs) to link these alterations to amplification or suppression of the influence of specific choice-relevant attributes on choices.”

“Neural encoding of choice attributes and effects of regulation

Next, we examined neural underpinnings of goal-consistent increases/decreases in the influence of attributes on altered choices in both tasks. […] We also explicitly tested whether the locus of effect differed across attributes (e.g. tastiness/healthiness, $Self/$Other/Fairness) or choice domain (e.g. social, non-social).”

“Individual differences in regulatory success

Are some people generally more successful using cognitive regulation of decision making or does it depend on the choice domain? […] It should also be predicted by neural activation in distinct, non-overlapping brain regions.”

C) Understanding of Outcomes.

We hope that the greater clarity of our framing and predictions now make clearer the implications of our results for understanding how regulation affects value representations at a mechanistic level. We have substantially revised the Discussion section to address the three questions we raise in the Introduction: 1) does cognitive regulation of decision making operate by changing attribute-level or integration-level representations? 2) Does the level at which it operates vary by domain? 3) What role do VMPFC and DLPFC play in this process?

Generally I think the question is very interesting and the approach is attractive, but I simply could not follow how the authors framed and then conducted their analysis. I will be interested to see what the other reviewers say. I might grasp things better if it were more clear how the theoretical concepts were operationalized for the analyses – precisely what is "regulatory" for example and how is it distinguished from non-regulatory versus just not involved by the task and then in the data analysis.

Again, we thank the reviewer for the positive assessment of the importance and interest value of this question. Please see our second response (above) for a more detailed discussion of the ways in which we have changed the framing, as well as discussion of the hypotheses and results, to make it clearer what we predicted and what we found, as well as how this relates questions of regulatory vs. non-regulatory regions.

We also thank the reviewer for bringing to our attention the lack of clarity in the description of our analyses (as did reviewer #2). In addition to revisions of the Results section, we have now also included a figure that further illustrates the key aspects of the multi-voxel decoding approach that we used to identify brain regions that differentially encode attribute values (or decision values) across goals/task conditions (Figure 4). We hope that this will make it easier for the broad and heterogeneous readership of *eLife* to follow the logic of this central analysis of our paper.

Reviewer #2:Tusche and Hutcherson present a thought-provoking and methodologically impressive study on cognitive regulation of dietary and altruistic choices, a topic of broad interest. The analyses, which combine a drift diffusion computational model of attribute-weighted choice and trial-by-trial MVPA decoding, are sophisticated, appropriate, and comprehensive. They show that while attribute values across choice and regulatory goal contexts can be decoded in VMPFC, they do not appear to be modulated by regulatory goal. Conversely, attribute values in DLPFC are modulated consistently with model-derived behavioral weights across regulatory goals that emphasize either healthiness or tastiness during dietary choice, and personal gain in altruistic choice contexts. Regulatory goals that emphasize another's benefits (e.g. feelings), however, could instead be decoded from right TPJ and precuneus, but not DLPFC, suggesting representations of others' wellbeing is modulated according to the prosocial regulatory goal but only when it requires theory of mind. These findings speak to both the domain-generality and domain-selectivity of cognitive regulation of decision making and importantly advance our understanding of the neural systems important for cognitive control and decision making more generally. In general I am supportive of this paper but think some outstanding issues could be better addressed to confirm some of their interpretations and rule out others.

We thank the reviewer for this positive evaluation of our study and the approach, as well as for the constructive comments and suggestions! We have performed several supplemental analyses in response to his/her suggestions, which have significantly strengthened the manuscript and provided further support for our interpretations. We detail these analyses below in our response to the reviewer’s specific comments.

It is notable that DLPFC flexibly encoded values of tastiness, healthiness, and $Self but not $Other or $Fairness on the one hand but strongly predicted altruistic choices on the other hand. Is the DLPFC prediction of altruistic choices then mediated especially by the change in $Self during altruistic choices, but not the change to $Other or $Fairness?

The reviewer raises an interesting question! We addressed this idea in the following way. First, at the behavioral level, we regressed changes in generosity (ΔGenerous Choices [(PC, EC) - NC]) on changes in model-based weights for subject’s own benefits (Δw $Self [NC - (EC, PC)]), and calculated the residuals. This gave us a measure of change in generosity after controlling for change in weight on $Self. Next, we repeated the cross‐subject decoding analysis (SVR) using neural response patterns in the DLPFC as features and the estimated residuals as labels for the prediction. This tells us whether the DLPFC continues to predict goal-consistent increases in generosity after controlling for change in the input of $Self on choices. Our results are consistent with the idea that the DLPFC predicts change in generosity because it encodes changes in weight on $Self: activation patterns in the DLPFC no longer predicted individuals’ goal-consistent changes in generosity when we controlled for changes in the weight of self-related benefits on choices (r=0.11, p=0.317, permutation test, [CI: -0.41, 0.41]). We next sought to show that this effect is specific to changes in self-related considerations. To do this, we repeated this supplemental analysis for each of the other choice attributes (i.e., regressing out change in weight on $Other or change in weight on Fairness, and then asking whether DLPFC predicts residual changes in generosity). We found that DLPFC prediction of altered generosity remained highly significant when we controlled for altered inputs of other’s benefits (r=0.51, p=0.016, [CI -0.41, 0.40]) or fairness considerations (r= 0.56, p=0.007, [CI -0.39, 0.39]) on choices. Taken together, these findings indicate that predictions of goal-consistent changes in generosity in the DLPFC are mediated specifically by altered weighs of subjects’ own monetary benefits on choices (Δw $Self). This finding is also consistent with evidence of a specific functional role of the DLPFC for altered representations for self-related considerations, while changes in attributes of $Other and Fairness are encoded elsewhere.

We now refer to this supplemental evidence in the Results section of the manuscript:

“Supplemental analyses suggest that predictive information on altered generosity was driven by neural information on changes in the attribute encoded in the DLPFC ($Self) and not by other attributes of the altruistic choice task (e.g., $Other, fairness) (see Appendix 1 – 1.5).”

We also present the details of these supplemental analyses in the Appendix. Due to the close match of this section with our description above, we refrain from re-iterating the details here and refer reviewer #2 to the relevant section of the supplemental material:

Appendix 1 – 1.5. DLPFC-based prediction of goal-consistent changes of generosity is driven by goal-consistent changes in attribute representations of $Self (but not $Other or Fairness).

A related question: Do the rTPJ and precuneus group effects predict individual differences in regulatory success during altruistic choices when goals depended on another's thoughts or feelings? Such evidence would tie together their argument that these latter regions "assume responsibility" for regulatory success when DLPFC does not because of the component process required to meet that goal.

We thank the reviewer for this suggestion. To address it, we ran supplemental decoding analyses to explicitly test if neural response patterns in the Precuneus and/or rTPJ predict individual differences in regulatory success in the altruism task. The analyses closely matched the prediction of regulatory success based on neural responses in the DLPFC with the following differences: First, ROIs were based on clusters in Precuneus and rTPJ in which information about other’s benefits ($Other) varied depending on the behavioral goal (for details see Table 4, for illustration see Figure 2H and Figure 2—figure supplement 1). Second, neural activation patterns for this prediction were extracted from subject-specific contrast images in GLM2 (altruism task) based on the contrast image of [PC > EC], since this was the contrast that yielded reliable differences in attribute representations for both ROIs (see Author response image 1 for illustration).

**Author response image 1. respfig1:** Multi-voxel response patterns in the precuneus reliably predicted individuals’ altered generosity in the altruism task (ΔGenerous Choices [PC – EC]: r = 0.57, p = 0.002 [CI: -0.41, 0.38]; ΔGenerous Choices [(NC, PC) – EC]: r = 0.61, p = 0.004 [CI: -0.41, 0.41]). This finding is consistent with the pattern of results on goal-dependent encoding of others benefits in this brain region (left panel). Response patterns in the rTPJ, on the other hand, did not significantly predict individual’s goal-dependent changes in generosity (all p’s ≥ 0.065).

This latter null result has to be interpreted with some caution however. Depending on details of the ROI selection and parameters of the decoding approach, we did observe significant predictions of regulatory success based on rTPJ responses. However, given the variability in these results, the lack of an independent replication sample to further validate the modulated decoding approach, and the complexity of the manuscript, we prefer not to expand on this TPJ finding in the manuscript. Should the reviewer or the editor feel that this complementary evidence is necessary to further validate our interpretation, we are happy to provide details of this complementary analysis that hints at prediction of regulatory success based on response patterns in the rTPJ.

Overall, these findings are in line with our interpretation that other regions such as the Precuneus might ‘assume responsibility’ for goal-consistent regulatory success when the DLPFC does not. We now refer to this supplemental evidence in the Results section of the revised manuscript.

Specifically we now report:

“Precuneus encodes individual differences in regulatory success in altruistic choice.

Strikingly, patterns in the DLPFC did not decode regulatory success for social attributes that were flexibly encoded in other regions of the brain (i.e., $Other, Fairness). […] We found that response patterns in the precuneus reliably predicted individuals’ altered generosity in the altruism task (ΔGenerous Choices [PC - EC]: r = 0.57, p = 0.002 [CI: -0.41, 0.38]; ΔGenerous Choices [(NC, PC) - EC]: r = 0.61, p = 0.004 [CI: -0.41, 0.41]), suggesting that domain-specific attribute coding contributes to individual differences in regulatory control.”

Given the prior literature on this topic, and in order to rule out a model whereby VMPFC value representations are modulated in an attribute-specific manner that depends on regulatory goals, it would be informative to see an ROI-based analysis of VMPFC. I may have missed something but I could not find one. The ROI from the conjunction presented in Figure 3 could be used or prior literature could be used.

Following the reviewers’ suggestion, we used an ROI-based approach to further examine if neural response patterns in the VMPFC encoded goal-consistent change in attribute representations. The ROI consisted of voxels in the VMPFC identified in the conjunction analyses (see Figure 3 for illustration, upper panel in Author response image 2). We tested this question in three different sets of analyses:

1) For the first approach, we examined average predictive information in the VMPFC-ROI identified by the whole brain searchlight decoding analyses described in the paper: For each participant and each condition, we extracted decoding accuracies for each voxel in the VMPFC-ROI and estimated an ROI-based average for each choice-attribute. Next, for each attribute, we ran a repeated measures ANOVA to explicitly test for altered information content. Confirming results from whole-brain analyses, information about the values of choice attributes (as measured by decoding accuracies) was not significantly modulated by the regulatory goal (all p’s > 0.13, uncorrected). This finding is in line with our whole-brain results (implemented in SPM), and suggests that the lack of results was not due to overly stringent statistical thresholds. This provides further support for our interpretation that value representations in the VMPFC on the attribute level are unaffected at a group level by our implemented regulatory goals.

2) However, this analysis leaves open the possibility that local response patterns in the VMPFC (rather than average decoding accuracies) might differentiate attributes by regulatory goal. Thus, we ran supplemental ROI-based within‐subject decoding analyses to predict trial-wise values of an attribute based on local response patterns in the VMPFC. These analyses were similar to the searchlight decoding approach with the difference that ROI-wise response patterns in VMPFC served as neural features for the predictions. Thus, instead of extracting multi-voxel patterns from a sphere around individual voxels in the ROI, we use one response pattern consisting of all voxels of the ROI to define neural response patterns. For each attribute separately, we estimated how well a subject’s multi-voxel response patterns encoded attribute values in a particular task condition. Next, on a group level, we used repeated measures ANOVA to examine whether neural information on an attribute varied depending on the behavioral goal. Consistent with the searchlight decoding approach, predictive information encoded in the VMPFC did not significantly vary cross task conditions (all p’s > 0.29, uncorrected).

3) Finally, we examined if response patterns in the VMPFC encoded goal-dependent changes across participants. Such a pattern might suggest that even though we did not observe main effects, VMPFC attribute representations still contribute in some way to regulatory success. Thus, in addition to the within-subject analyses described above (1-2), we also ran cross-subject decoding analyses. These analyses were identical to the decoding analyses of individual regulatory success from neural response patterns of the DLPFC, with the exception that response patterns were extracted from the VMPFC cluster identified in the conjunction analysis. Consistent with the lack of main effects for the supplemental analyses reported above, cross-subject decoding analyses did not yield significant predictions of participants’ regulatory success for any attribute or for choice behavior (i.e. healthy or generous choices) in the food task or altruism task (all p’s > 0.31, uncorrected).

Finally, to address potential concerns that the selected ROI might have been too small for the multi-voxel analyses approach, we also repeated these analyses (1-3) using a larger VMPFC‐ROI (see Author response image 2, lower panel). The original conjunction area in the VMPFC (upper panel) represents only the central voxels of the searchlight spheres that encoded all attribute values in the attribute-specific searchlight decoding analyses. The lower panel illustrates the VMPFC-ROI based on all voxels within spheres that predicted all attribute values. Results of the analyses (1-3) matched those for the original VMPFC ROI, suggesting that null-findings in supplemental analyses are not merely driven by the comparatively small number of voxels identified in the conjunction analyses (Tastiness, Healthiness, $Self, $ Other, Fairness).

**Author response image 2. respfig2:** The figure illustrates two VMPFC-ROIs used for the supplemental analyses.

Taken together, these more sensitive and fine-grained ROI analyses confirm our conclusion that, in this study, our VMPFC ROI encodes the values of each of the individual attributes, but not in a manner that is sensitive to regulatory goals. We have highlighted these additional analyses briefly in the Results section of the main paper, and expanded on them in the Appendix.

The relevant paragraphs in the revised Results section now reads:

“No evidence for goal-dependent coding of attribute values and decision values in the VMPFC. […] While activation patterns in the VMPFC (as well as several other regions) reliably predicted overall decision values in both tasks, regulation failed to modulate decoding accuracies for decision value (Supplementary file 2) or for any specific attribute (Appendix 1 – 1.3), and did not predict individual differences in regulatory success (Appendix 1 – 1.3).”

“VMPFC does not encode individual differences in regulatory success.

Because of its hypothesized role in valuation, a post-hoc analyses also examined whether the VMPFC region that encoded all attributes predicted regulatory success in either choice task. […] However, exploratory functional connectivity analyses provided subtle hints that the VMPFC could be indirectly related to regulatory success through its modulation of both DLPFC and precuneus (see Figure 3—figure supplement 2 and Appendix 1 – 1.6 for details).”

For detailed descriptions (matching those presented above) we refer reviewer #2 to the respective section of the Appendix:

“Appendix 1 – 1.3 ROI-based post-hoc tests to identify goal-consistent value coding in the VMPFC.”

One question which is unclear from the conjunction analyses is whether it is the same neural code (e.g. in DLPFC) that is used across task contexts, or whether the code is distinct (context-dependent) but found in the same brain region. What do the authors find if the SVR is trained on one attribute (e.g. tastiness) and tested on the other two (e.g. healthiness and $self)?

The question of how the DLPFC encodes different attributes is extremely interesting! The reviewer is also correct that the main analyses of the manuscript do not address this issue. We therefore ran a supplemental decoding analysis to explicitly test if neural codes in the DLPFC are similar or distinct across contexts, attributes, and tasks. More precisely, we tested if a decoding model trained on neural data in one context (or attribute, or task) allows decoding the attribute values in a different context (or attribute, or task) based on neural data observed in these trials. Significant decoding would suggest that the neural code in one context is reinstated (at least in part) in the different context.

To do this, we used a post-hoc ROI-based approach using voxels in the DLPFC (Figure 5A) together with a within‐subject decoding approach (as described in the manuscript). However, instead of a leave-one-run-out cross-validation, we now trained the SVR model on data of one condition (e.g. Tastiness-HC) and tested it on data of another condition (e.g. Tastiness-TC). The procedure was then repeated, reversing training and testing data (e.g. train on Tastiness-TC, test on Tastiness‐HC). Decoding accuracies were estimated by averaging predictions across this 2-fold cross-validation. As a sanity check, we also ran the analysis for data within a condition (e.g. train on 1st half of data in Tastiness-HC, test on 2nd half of data in Tastiness-HC, and vice versa). Statistical significance was assessed at the group level using permutation tests in which we examined how likely observed decoding accuracies were compared to an empirical null-distribution of these predictions. Null-distributions were estimated by breaking up the matching of trial-wise neural activation patterns and attribute values (1000 folds). Upper cutoff-values for significance were the 95th percentile of null-distributions.

Our results suggest that the DLPFC maintains distinct neural codes for different attributes, but that such codes may be similar across different contexts. Figure 5B illustrates the results of this analyses for attributes encoded in the DLPFC (Tastiness, Healthiness, $Self).

First, we examined whether values of a particular attribute (e.g. Tastiness) relied on common neural codes across contexts/regulatory conditions (within-attribute decoding, yellow frames). For each attribute, we found evidence in favor of common coding of attribute values across contexts (illustrated in red, signifying significant cross-condition classification, p < 0.05 in stringent permutation tests). For example, similar neural codes seem to represent values of Tastiness in TC and HC. However, a lack of significant cross-condition classification in 2 of the 18 tests (e.g. Tastiness in NC and HC) might also point to some degree of context-dependent differences in activation patterns for attribute values, or to the much weaker representation of some attributes in some conditions.

Second, and more importantly, the analyses enabled us to explicitly test whether neural codes in the DLPFC generalize across attributes (cross-attribute decoding). Response patterns that coded for values of one attribute did not allow predicting values of another attribute (illustrated in blue). This was found to hold true across regulatory conditions. This finding suggests that distinct neural codes within the DLPFC are used to encode values of individual attributes in both choice tasks (Tastiness, Healthiness, and $Self).

We now report these findings in the revised Results sections of the manuscript:

“This finding suggests that the DLPFC acts as a domain-general circuit for goal-sensitive value representations. But what does this convergence in the DLPFC signify? […] This supports the idea that the DLPFC acts as a domain-general mechanism for representing different attributes in a goal-sensitive manner, using unique codes for each attribute.”

We also refer to these findings in the revised Discussion section of the manuscript:

“Third, although we observed converging effects of regulation for a subset of attributes in the DLPFC (including tastiness, healthiness, and self-related benefits), representations of these attributes utilized distinct, differentiated codes. Taken together, although our results do not preclude the possibility that in other contexts cognitive regulation of decision making might operate on a single, centralized value integration mechanism, they suggest that it may often operate by changing distinct attribute representations.”

We believe that this set of supplemental analyses has significantly strengthened the manuscript and has shed further light on the neural computations in the DLPFC that underlie cognitive regulation of value-based choice.

Can the overlap between representations be better visualized?It would be informative to visualize the feature weights for the key areas (e.g. VMPFC, DLPFC) across voxel space in every subject. This procedure should help to assess to what extent any decoding effects are due to hard anatomical boundaries between subareas (e.g. dorsal and ventral aspects of DLPFC) or to distributed patterns within areas.

The reviewer makes an interesting (but challenging!) suggestion. Using a post-hoc ROI approach based on voxels in the DLPFC (Figure 5B), we re-ran the within-subject decoding analyses of values of tastiness, healthiness, and $Self (separately for each condition). We then extracted feature weights of the prediction for each of the attributes encoded in the DLPFC, subject, and condition and projected them back into MNI space (3 attributes x 36 subjects x 3 conditions = 324 images). Visual inspection of feature weights did not reveal systematic clustering of feature weights across voxel space or hard anatomical boundaries. Because of the large amount of data, we have opted for now to leave out a presentation of every attribute in every subject and condition, as suggested by the reviewer. The full set of the 324 images of brain images in the MNI space (nifti format) can be found here: https://www.dropbox.com/sh/7s7edpdvq9zjgmd/AADDm1SC8Y_yPGi0bCH18kuFa?dl=0

See Author response image 3 for an illustration of the results for one representative subject (subject 1, [MNI 45, 38, 22], xjview (http://www.alivelearn.net/xjview/)).

**Author response image 3. respfig3:** 

In addition, to provide the reviewer with a summary of the extracted feature weights, we have prepared several author response images. For each subject (s1 to s36), we plotted feature weights (color-coded) of all voxels in the DLPFC-‐ROI (vox) for Tastiness, Healthiness and $Self (columns 1-3). Separate figures were created for task-conditions: NC (Author response image 4), HC and EC (Author response image 5), and TC and PC (Author response image 6). As noted above, visual inspection of feature weights did not reveal hard anatomical boundaries (i.e., systematic clustering of voxels with high positive/negative weights in the SVM model as indicated in darker red and/or blue) but rather point towards distributed coding for all three attributes encoded in the DLPFC.

**Author response image 4. respfig4:** Voxel-wise feature weights of the Support Vector Regression Model in Natural trials [NC] in both choice tasks.

**Author response image 5. respfig5:** Voxel-wise feature weights of the Support Vector Regression Model in Health [HC] and Ethics trials [EC].

**Author response image 6. respfig6:** Voxel-wise feature weights of the Support Vector Regression Model in Taste [HC] and Partner trials [PC].

We also note that the current manuscript focuses on brain regions that show goal-dependent coding of attribute values. The issue of predictive features across multiple attributes in the VMPFC will be reported elsewhere in greater detail. The latter touches on the popular notion of so called ‘common currency’ coding in value integration areas of the brain that will be presented together with data of another fMRI task not included in the current manuscript.

How do the authors interpret the altered representations in some primary sensory and motor areas, e.g. less strong decoding of tastiness values for NC than HC and TC? Have the authors considered a model whereby coupling between DLPFC and distinct regions of sensory cortex is modulated according to the regulatory goal? The DLPFC must get its attribute representations from someplace.

We speculate that part of the altered representations in visual and motor cortex may be due to differences in patterns of eye movements, visual attention, and the vigor of motor responding. While these effects are not uninteresting, we suspect they are a corollary of changes happening in other areas, rather than drivers of regulation per se.

We wholeheartedly agree with the reviewer that, in theory, information transfer between the DLPFC and other areas could account for changes in attribute representations. However, it is not entirely clear what the appropriate method is to test for this coupling, given that attribute encoding effects in the DLPFC appeared only in multivariate response patterns, not univariate response, while current functional connectivity methods assume that connectivity will manifest in the coupling of univariate responses in one region to univariate responses in another. We suspect that functionally coupling actually occurs at the level of distributed representations within and across regions, which complicates the interpretation of effects in univariate connectivity analyses.

Nevertheless, and with that caveat in mind, we performed a complete set of connectivity analyses for four regions of interest (ROIs) identified in our dataset: 1) the DLPFC area showing a conjunction between regulatory effects on representations of Tastiness, Healthiness, and $Self, 2) the TPJ and 3) the precuneus showing altered attribute encoding for $Other, and 4) the VMPFC area showing a conjunction across all attribute representations. For each of these regions, we performed a beta series functional connectivity analysis (Rissman et al., 2004), a method that has more power to detect context-specific changes in functional connectivity for event-related designs (Cisler et al., 2014). These supplemental analyses explicitly tested for changes in functional connectivity that might explain altered representations in these regions:

1) Based on the reviewers’ comment, we first tested the possibility that DLPFC might be more connected with visual or motor areas during regulation, either on average, or as a function of regulatory success. To this end, we defined the DLPFC as seed region, extracted trial-by-trial responses from this region, and then used these responses as parametric modulators in a GLM to identify areas where response covaried with DLPFC differently as a function of regulation. These analyses did not reveal goal-dependent functional connectivity with either primary visual or motor areas that showed goal-dependent attribute coding, even at liberal statistical thresholds (p < 0.005, uncorrected).

2) Following up on the notion that the DLPFC must get its attribute representations from someplace, we then tested the possibility of functional coupling between the DLPFC and the VMPFC region shown to encode all attributes in a goal-independent manner. We observed the following patterns:

a) During food choices, increased coupling between VMPFC and DLPFC during Health vs. Natural trials [HC > NC] reflected the extent to which participants increased the weight of healthiness in Health vs. Natural trials (Δw Healthiness [HC > NC], p < 0.005, uncorrected).

b) During altruistic choices, decreased coupling between VMPFC and DLPFC during Natural vs. Regulatory trials [NC > (EC, PC)] predicted the extent to which the weight on $Self decreased in Natural vs. Regulatory trials (Δw $Self [NC > (EC, PC)], p < 0.005, uncorrected).

These results are detailed in the revised manuscript and the Appendix:

“Figure 3—figure supplement 2. Exploratory functional connectivity analyses.

A. Region of the VMPFC (red) where increased connectivity with the DLPFC during Health vs. Natural and Taste focus conditions correlates with Δw Healthiness in Health vs. Natural and Taste Focus. […] All results are shown thresholded at p < 0.005 uncorrected.”

“Appendix 1 – 1.6 Changes in functional connectivity with the VMPFC correlate with regulatory success”.

In addition, we have now included a brief reference to this issue in the revised Results section and Discussion section of the manuscript.

“However, exploratory functional connectivity analyses provided subtle hints that the VMPFC could be indirectly related to regulatory success through its modulation of both DLPFC and precuneus (see Figure 3—figure supplement 2 and Appendix 1 – 1.6 for details).”

“This raises an important question: what determines the capacity of the DLPFC to properly represent these different attributes? […] Future work including the use of measures with higher temporal precision may help to elucidate when and how interactions between the VMPFC and DLPFC determine regulatory success in different contexts.”

The one comparison in which a match with the behavioral analyses failed to emerge was for ethical considerations compared to normal or personal considerations. I do not view this null finding as problematic, but out of curiosity do the authors have any thoughts as to what is going on when regulatory success depended on changes to fairness due to the goal of complying with social norms? This null finding should be addressed in the Discussion.

We fully share the reviewer’s curiosity about the meaning of this result! We think there are two possibilities here that are not mutually exclusive. First, we speculate that the instructions to focus on ethical reasons for giving are less constrained than instructions to focus on one’s partner’s feelings. Ethical values can be derived from many sources (including fairness, deontological views about the correct action, consequentialist views about the correct outcome, etc.). Prior research suggests that there is considerable individual heterogeneity in which constructs people deem relevant, and we suspect that is in part what is happening here. In contrast, focusing on one’s partner’s feelings may result in more consistent focus on specific considerations and attributes related to empathy and perspective taking across subjects.

However, we also believe our results are just as consistent with a second interpretation: that the lack of a correspondence between behavioral weights and neural response in the ethics condition speaks to the idea that the behavioral weights that we estimate, despite giving a single number for a specific attribute, may actually result from the consideration of multiple sub-attributes served by specific computations being performed in specific regions. In other words, the increased weight on $Other in both Partner and Ethics focus conditions may reflect empathetic considerations of consequences in the Partner condition, but may reflect consideration of norms and rules (e.g., do unto others as you would have done unto you) in the Ethics condition. That neural activation patterns in the TPJ and precuneus encode $Other in the Partner focus condition and not in the Ethics condition suggests that their computations may support the former more directly than the latter.

We have revised the Discussion section to focus on this second possibility, specifically considering its implications for the computations occurring in the TPJ and precuneus:

“At the same time, goal-consistent changes in pro-social attributes (e.g. others benefits) appeared in areas like the TPJ and precuneus, especially when focused on the partner’s thoughts and feelings. […] Thus, the TPJ and precuneus appear to encode features specifically related to representing others’ outcomes in a goal-sensitive manner, pointing to specialized loci of cognitive regulation in social choice domains.”

We have opted not to include the first point (heterogeneity of strategies) because the Discussion is already a bit long, and we feel that other points are more critical to make. However, if the reviewer or editor feels that a discussion of this point is worth mentioning, we are of course very happy to include it.

Reviewer #3:In this manuscript, Tusche and Hutcherson report an fMRI study on how cognitive regulation affects the neural representation of choice-relevant attributes. They look for mechanisms that may generalize across two types of choices, one involving conflict between healthiness and tastiness of food items, the other involving conflict between self-interests and altruistic concerns. In different conditions, participants are asked to focus on one or the other attribute, which regulates the weights assigned to the targeted attributes, as shown via computational modeling of choice data. The key findings are the links between these changes in attribute weights and the decoding accuracy obtained for these attributes using multivariate pattern analysis (MVPA) in various cortical regions. The results are quite convincing, with successful decoding across tasks and individuals. There is no clear conclusion about whether the regulation is centralized or distributed though, since changes in decoding accuracy are observed in the DLPFC for most of the attributes but not all.

We fully concur that the DLPFC finding is convincing and intriguing, but not entirely consistent with either a fully centralized or fully distributed model. Response patterns in the DLPFC encoded values of three choice attributes (tastiness, healthiness, $Self) in a goal-consistent manner, while attribute values of Fairness and $Other were flexibly encoded in attribute-specific brain regions. Although this could be seen as providing “no clear conclusion” we believe our data, especially with the addition of new analyses suggested by our three reviewers, allows us to make a number of important and clear inferences. Specifically, we believe the paper presents convincing evidence in favor of a “hybrid model” in which goal-dependent changes in attribute values occur at the attribute level in attribute-specific loci for some attributes, but in a common hub in the DLPFC for other attributes. We believe that this is an important finding in its own right that will inspire future research on adaptive decision making. We have substantially modified the Introduction and Discussion to further clarify this point. More importantly, we now also report additional analyses of DLPFC responses that suggest that this common hub flexibly represents distinct attributes using distinct neural codes (as opposed to a unified value signal that generalizes across attributes and domains), shedding more light about how the DLPFC encodes goal-dependent attribute values. We now refer to these additional analyses and findings (suggested by reviewer #2) in the revised Results section of the manuscript:

“This finding suggests that the DLPFC acts as a domain-general circuit for goal-sensitive value representations. […] This supports the idea that the DLPFC acts as a domain-‐general mechanism for representing different attributes in a goal-‐ sensitive manner, using unique codes for each attribute.”

“Figure 5. Domain-general locus of goal-dependent attribute coding. […] This pattern of results indicates that goal-sensitive representations of attribute values in DLPFC rely on attribute-specific neural codes.”

Taken together, the sophisticated analyses and novel results significantly improve our understanding of when and why an individual will show specific or more global deficits in regulatory success in dietary and social choice, making our paper of high interest to the broad readership of *eLife*. Moreover, they open up several novel and interesting questions. The most prominent one is certainly the following question: what distinguishes attributes whose representations converge in DLPFC from those that did not? Are attributes that are flexibly encoded in the DLPFC simply less social in nature, less abstract and thus easier/faster to construct, or do they influence outcomes for the participant more directly? Identifying the precise factors that determine whether and when the DLPFC acts as the site for cognitive regulation of value opens up exciting new avenues for future research. We believe that the revised discussion of centralized and distributed loci of goal-consistent attribute coding has significantly strengthened the manuscript.

The role of cognitive control in economic choice is poorly understood and this study brings valuable insights by applying MVPA to standard choice paradigms. My main concern is the absence of a mechanistic account linking brain activity to behavioral output. In a sense, the results seem a bit trivial: when participants pay attention to a particular type of information (by instruction), this information is easier to decode from their brain activity. This effect should be seen as the outcome (and not the process) of cognitive regulation. How cognitive regulation is implemented, and how better decoding translates into biased choices, still need to be explained.

We thank the reviewer for his/her critical comments and suggestions, and for the assessment that this study yields valuable insights! We agree with the reviewer that the following assumption is extremely intuitive: when participants pay attention to a particular type of information (by instruction), this information is easier to decode from their brain activity. However, for several reasons we believe that evidence in support of this notion is by no means trivial and that our findings provide valuable insights into the mechanisms that underlie cognitive regulation of value in the brain.

First, as we make clear in the revised Introduction and Discussion sections, extant research has actually not consistently found such patterns despite clear changes in behavior, raising an important puzzle to be explained. Our results are the first to identify a clear representation of all choice-relevant attributes across multiple contexts, and to show clear modulation of all choice-relevant attributes in a manner depending on regulatory goals. Second, we also think our specific pattern of results clearly speak against the trivial interpretation that one would have to observe changes in attribute coding in the brain, and that this is an uninteresting reflection of changes in behavior. We observe a highly specific pattern of altered and unaltered representations of attribute values: for each attribute, we identified brain regions that reliably encoded current attribute values independent of regulatory condition. The VMPFC is one example for this finding, highlighted in the manuscript. In fact, a lack of modulation was actually the rule rather than the exception. Only a limited number of brain regions contained information on attribute values that varied significantly across task conditions. More importantly, we find a coherent set of diverging and converging changes in in attribute representation: while some attribute changes can be decoded from the DLPFC, changes in social attribute representation appear specifically in social cognitive areas. This is anything but trivial, and, we believe opens up crucially important lines of questioning that will be the basis for a better understanding of cognitive regulation and its mechanistic implementation. We have substantially revised the Introduction and Discussion to highlight these important points.

Third, we completely agree with the reviewer that our results focus more on outcome (and not the process) of cognitive regulation. We have substantially revised the Introduction to clarify this point further. However, we respectfully disagree that no information about process or mechanism can be derived from a detailed description of where such changes are observed. We have tried to make this clearer in the following ways:

1) We have now clarified in the new version of the paper in the Introduction, Results, and Discussion how identifying where attribute encoding changes occur may allow us to distinguish between different theories about computational process and mechanism. Specifically, we make clearer that regulation could occur via two processes: modulation at the level of specific attribute representations in non-overlapping, attribute-specific regions, or at the level of weighting in the value integration process. We show how our results are clearly consistent with attribute-level modulation for some attributes (i.e., $Other in TPJ and precuneus, fairness in middle frontal gyrus), and how our results are more ambiguous for other attributes (i.e., modulation of Tastiness, Healthiness, and $Self in DLPFC).

2) We also make clearer that there are important outstanding questions about the domain-generality or domain-specificity of regulation effects. On the one hand, if different attributes are computed in different regions with different computational properties, regulation could operate through different channels in different domains. If, instead, regulation operates at a more domain-general level, then regulation might influence attribute representations in a common region across multiple different attributes and multiple choice contexts. Understanding whether regulation changes attribute representations in a domain-general or domain-specific way can help reveal both the targets of regulation, as well as the mechanism of action.

In the Discussion, we also highlight how these questions are not of purely theoretical importance: the locus of action (attribute or integration level, domain specific or domain general) predicts non-trivial differences in patterns of regulatory success, which we observe in our own data. For attributes where regulation influences representation in a common region (i.e., DLPFC) regulatory success tracks together, but for attributes represented elsewhere (i.e., $Other in TPJ and precuneus), regulatory success appears to operate independently. Our results also may help to explain when and why regulatory success might feel difficult: if, as we observe, there are many regions of the brain where attribute representations are not altered by regulation, then these altered representations may continue to leak into the choice process and interfere with regulatory success.

3) Finally, we have included several supplementary analyses, including analyses that tested for commonalities of goal-consistent attribute coding in the DLPFC as well as functional connectivity results, which shed some light on how these computational processes are implemented in the brain. Please see our detailed first response to reviewer 3. above (neural attribute coding in the DLPFC across attributes and tasks) and our response to the next point below for more information (exploratory functional connectivity).

Taken together, we believe that these revisions have significantly strengthened the manuscript and provide valuable insights into neural computations in cognitive regulation of decision making across contexts and domains.

For the latter point, it would be important to show that the decoded value (i.e., the decoder output) correlates with the behavioral weights. The alternative would be that changes in decoding accuracy correspond to changes in precision (i.e., signal-to-noise ratio) and not changes in the signal itself. If correct, this would mean that a region downstream to the DLPFC could just read this value, add it to other values corresponding to other attributes, and feed the aggregate value to a selection process that makes the decision. Perhaps functional connectivity could be used to test for such a transfer of information. Thus, the neural model would parallel the behavioral model.

The reviewer suggests two supplemental analyses. First, we examined the link of decoding accuracy and estimated behavioral weights. To this end we ran supplemental regression analyses. Please see our sixth response to reviewer 3 below for details.

Second, we wholeheartedly agree with the reviewer that, in theory, information transfer between the DLPFC and other areas could account for changes in attribute representations. However, it is not entirely clear what the appropriate method is to test for this coupling, given that attribute encoding effects in the DLPFC appeared only in multivariate response patterns, not univariate response, while current functional connectivity methods assume that connectivity will manifest in the coupling of univariate responses in one region to univariate responses in another. We suspect that functionally coupling actually occurs at the level of distributed representations within and across regions, which complicates the interpretation of effects in univariate connectivity analyses.

Nevertheless, and with that caveat in mind, we performed a complete set of connectivity analyses for four regions of interest (ROIs) identified in our dataset (as also suggested by reviewer 2, see our seventh response to reviewer 2): 1) the DLPFC area showing a conjunction between regulatory effects on representations of Tastiness, Healthiness, and $Self, 2) the TPJ and 3) the precuneus showing altered attribute encoding for $Other, and 4) the VMPFC area showing a conjunction across all attribute representations. For each of these regions, we performed a beta series functional connectivity analysis (Rissman et al., 2004), a method suggested to have more power to detect context-specific changes in functional connectivity for event-related designs (Cisler et al., 2014). These supplemental analyses explicitly tested for changes in functional connectivity that might explain altered representations in these regions:

1) Using the DLPFC conjunction area as seed region, we first tested the possibility of functional coupling between the DLPFC and the VMPFC region shown to encode all attributes in a goal-independent manner would change either overall, or as a function of regulatory success. Consistent with this notion, we observed the following patterns:

During food choices, increased coupling between VMPFC and DLPFC during Health vs. Natural trials [HC > NC] reflected the extent to which participants increased the weight of healthiness in Health vs. Natural trials (Δw Healthiness [HC > NC], p < 0.005, uncorrected).

During altruistic choices, decreased coupling between VMPFC and DLPFC during Natural vs. Regulatory trials [NC > (EC, PC)] predicted the extent to which the weight on $Self decreased in Natural vs. Regulatory trials (Δw $Self [NC > (EC, PC)], p < 0.005, uncorrected).

2) We also examined coupling using Precuneus and TPJ as seed regions. This analysis suggested that, during altruistic choices, increased coupling between the VMPFC and Precuneus on Partner vs. Ethics trials also correlated with greater weighting of $Other in Partner vs. Ethics trials (Δw $Other [PC > EC], p < 0.005, uncorrected).

3) Complementary analyses using the VMPFC as a seed region yielded similar patterns.

Our results thus suggest precisely the relationship that the reviewer is suggesting: the DLPFC, TPJ, and Precuneus may be receiving information from – or sending information to – the VMPFC. We now report results for these exploratory, complementary connectivity analyses in the revised version of the manuscript and the Appendix:

“Figure 3—figure supplement 2. Exploratory functional connectivity analyses.[…] All results are shown thresholded at p < 0.005 uncorrected.”

Detailed descriptions of the methods and results of this exploratory functional connectivity analyses can be found here:

“Appendix 1 – 1.6 Changes in functional connectivity with the VMPFC correlate with regulatory success”.

In addition, we refer to these exploratory findings in the Results section and Discussion section of the revised manuscript. Notably the section also acknowledges another reviewer comment (see our fifth response to reviewer

3 below) as well as the issue of directionality for the interpretation of functional coupling between brain regions (following up on comments of reviewer #2):

“However, exploratory functional connectivity analyses provided subtle hints that the VMPFC could be indirectly related to regulatory success through its modulation of both DLPFC and precuneus (see Figure 3—figure supplement 2 and Appendix 1 – 1.6 for details).”

“This raises an important question: what determines the capacity of the DLPFC to properly represent these different attributes? […] Future work including the use of measures with higher temporal precision may help to elucidate when and how interactions between the VMPFC and DLPFC determine regulatory success in different contexts.”

Other points:- The correlation across individuals could reflect compliance to the instructions rather than self-regulation capacity. The arguments taken from subjective report and from body-mass index are quite weak. For subjective report it could be that the rating scale is not reflecting the propensity to comply with the instructions. For body-mass index the opposite correlation could be expected: those who regulates food intake in real life should not need instructions in the lab.

We agree with the reviewer that compliance with reviewer instructions could be driving these results, either in part or in whole. This criticism can be leveled at the majority of studies on cognitive regulation, in both the decision making and emotion-regulation literature. However, we believe the specific pattern of results that we find argues against the simplest version of an experimental demand explanation. Specifically, we hypothesize that the motivation to comply with instructions likely affects both the motivation to suppress concerns about your own payoff during altruistic choices as well as increase those for another person (if not more so, as regulatory goals in the PC Partner Condition explicitly focused on $Other). However, behavioral evidence specifically links the ability to decrease self-related monetary consideration ($Self) to goal-consistent changes of food attributes (but not changes in $Other or Fairness), a finding mirrored in identified neural substrates of goal-consistent attribute coding.

However, the reviewer makes a valid point and we agree that we cannot fully distinguish between capacity and motivation explanations in this dataset. We now explicitly acknowledge this limitation in the revised Discussion section of the manuscript:

“We cannot completely rule out that regulatory effects on behavior and attribute representations might partly reflect differences in motivation to satisfy expectations of the experimenter. […] Tying laboratory measures of regulation to real-world consequences also remains a necessary future step in understanding the significance of these findings.”

We are also happy to include any specific reviewer suggestions in this paragraph on how to further reduce motivational or experimenter demand effects in both choice paradigms, in addition to our probabilistic implementation of choices (ensuring anonymity with regard to the randomly drawn and implemented trial at the end of both tasks, as both the experimenter and the partner are unaware if the outcome was based on a choice or a choice reversal), collected self-report measures of subjects’ motivation to comply with task instructions, and collected real-‐world measures of dietary success (BMI). These concrete suggestions might benefit future studies that aim to further elucidate the neural basis of cognitive regulation of decision making.

We also would like to briefly mention evidence on structural brain correlates of dietary regulatory success reported elsewhere (Schmidt et al., in press), because we believe these results may in part address the reviewer’s concern. Using voxel-based morphometry, this paper combines four independent datasets that examine goal-dependent changes in two different forms of dietary self-regulation. Note that in this independent paper, dietary success refers to altered weights of food attributes as estimated in a behavioral regression model (instead of a drift diffusion model as used in the current paper). We show that individual differences in dietary success correlate with differences in grey matter volume in the DLPFC, identified in a whole-brain regression analyses without any a priori assumption about regions of interests. Importantly, this localized neuronatomical correlate of dietary regulatory success matches our functional DLPFC ROI. See Author response image 7 for illustration. This finding indicates that goal-consistent changes in dietary choice at the behavioral level are likely not due solely to situational motivational effects, but might rather reflect more stable individual differences (as reflected in neuroanatomical markers in key regions of regulatory control). Upon acceptance of this manuscript, we intend to refer to this evidence in the respective Discussion section (see above) as additional reference.

**Author response image 7. respfig7:** 

Finally, we have taken the comment of reviewer #2 to heart and have removed the description of the BMI related evidence from the revised manuscript and have moved the description of self-report data on subjects’ motivation to comply with experimental instructions to the Appendix:

“Appendix 1 – 1.8 Self-reported motivation to comply with instructions and observed regulation- success.”

- The observation that all attributes are represented in the VMPFC but inaccessible to cognitive regulation is super interesting (and novel, to my knowledge). The dissociation with DLPFC should be more emphasized and discussed. Would this mean that VMPFC representations are closer to stimuli and DLPFC to responses?

We wholeheartedly agree with the reviewer about the novelty and importance of our VMPFC findings. Although there are hints in previous literature that the VMPFC may not always be the direct precursor of regulatory success, our study confirms and extends these earlier results in important ways. We have now performed a considerably more extensive analysis of VMPFC responses, using a post-hoc ROI-based approach as suggested by reviewer #2, confirming that decoding of attribute values in VMPFC doesn’t vary across regulatory goals and also fails to predict individual differences in success (in contrast to DLPFC, which does both). We have revised both the Introduction and Discussion sections to highlight the importance of understanding the roles of VMPFC and DLPFC in enabling regulatory success, and discuss the significance of the VMPFC null result more carefully in the Discussion section, incorporating the reviewer’s idea that the VMPFC may represent an earlier stage in the process of value construction:

The relevant sections in the revised Discussion section now read:

“Unexpectedly, cognitive regulation of decision making did not reliably modulate value signals within the VMPFC. Instead, regulatory effects converged to modulate a subset of distinct attribute representations in both the social and non‐social domain within a region of the DLPFC that has previously been implicated in value-based choice (Plassmann et al., 2007; Plassmann et al., 2010; Hutcherson et al., 2015)”

“The role of VMPFC and DLPFC in valuation and cognitive regulation.

Our study adds to a growing body of experimental work finding that behavioral effects of regulation can occur in the absence of corresponding changes to either overall levels of VMPFC response (Hollmann et al., 2012; Hutcherson et al., 2012; Yokum and Stice, 2013), or VMPFC representation of specific attributes like taste (Hare et al., 2011a). […] Future work including the use of measures with higher temporal precision may help to elucidate when and how interactions between the VMPFC and DLPFC determine regulatory success in different contexts.”

We also highlight the potentially distinct role of VMPFC and DLPFC in goal-consistent value coding in a separate goal of the study in the revised Introduction of the manuscript (Goal 3):

“Finally, we sought to shed light on whether information represented in VMPFC and dorsolateral prefrontal cortex (DLPFC) supports either attribute-level or integration-level changes in value during cognitive regulation of decision making. […] However, several failures to observe changes in the VMPFC during cognitive regulation of decision making (Hollmann et al., 2012; Hutcherson et al., 2012; Yokum and Stice, 2013) suggest the need to either measure value computation in a more sensitive way, or to identify alternate routes to behavioral change.”

- To compare the pattern of attribute weights and the pattern of decoding accuracy across conditions, the authors intend to reproduce significance of pair-wise comparisons. As they know this approach heavily depends on the statistical threshold, which may be matter of debate when multiple comparisons are made. I would favor a straight regression of decoding accuracy against weight (across conditions).

The reviewer raises an interesting point. A regression of decoding accuracy against weight (across conditions) allows testing whether the amount of local neural information on attribute values reflects individual differences in the behavioral weights of this attribute. We expected that this alternative analysis approach would confirm our main findings for attributes for which neural evidence closely matched predictions of the behavioral computational models (e.g. for taste and health). However, one major potential concern with the suggested regression approach occurs in cases in which neural evidence diverges from predictions of the behavioral model (such as for $Others in the rTPJ and Precuneus). For those key findings of our study, estimated behavioral weights (specifically for the ethics condition) clearly don’t match the neural substrate for goal-dependent coding of attribute values. Thus, while our repeated measures ANOVAs identify any brain region in which information content on attribute values varies across condition (independent of whether it is consistent with predictions of the behavioral model or not), the regression approach identifies only brain regions in which neural decoding accuracies fully and closely match predictions of the behavioral model. Thus, if the behavioral model is even slightly miss-specified, or if more than one region contributes independently to the single weight estimated by the behavioral model, the neural results could fail to conform to predictions. While we are sensitive to potential concerns related to the issue of multiple comparisons, we respectfully argue that our original analysis approach is more appropriate to study goal-dependent changes across a wide variety of attributes, based on these theoretical considerations.

This being said however, we agree with the reviewer that it is important to know whether decoding accuracies also match the behaviorally estimated weights. We ran several supplemental analyses to provide further evidence on our key findings (also addressing reviewer 3’s third comment). More precisely, following the suggestion of reviewer #3, we used straight regression analyses of decoding accuracies against weights (across conditions). This set of complimentary analyses adopted a whole-brain approach that examined whether behavioral attribute weights covary with local neural decoding accuracies using parametric regression analyses as implemented in SPM: For each attribute (e.g. for the neural coding of tastiness values), these group-level analyses estimated a general linear model in SPM (simple one-sample t-test against 0) that used whole-brain decoding accuracy maps of each subject (collapsed across conditions) as the dependent variable and the attribute weights as the predictor variable. Contrast images of the parametric regressor against implicit baseline identify brain regions in which decoding accuracies covary linearly with behavioral attribute weights.

First, we examined attributes in the food choice task. Confirming results of our main analyses, the whole-brain analyses found that neural decoding accuracies on healthiness values in the right DLPFC were modulated by the weight of healthiness on choices as estimated in the behavioral model (p<0.001, cluster-level corrected at p<0.05 FWE). Likewise, in line with our previous results, we found that decoding accuracies on tastiness values in the right DLPFC (peak in the right superior frontal gyrus, SFG) covaried with the weight that this attribute had in food choices (p<0.001, cluster-level corrected at p<0.05 FWE). No brain region showed a negative correlation of decoding accuracies and behavioral attribute weights. See Table below for details on the clusters.

Author response table 1: Brain regions in which neural decoding accuracies covaried with individuals attribute weights in the food task across conditions.

AttributeBrain regionSideTkMNI xyz**Healthiness**
Positive(D)LPFCR4.37133512328Visual CortexL/R6.461346‐3‐767CerebellumL/R4.841479‐37‐38NegativeMotor Cortex ‐L4.65217‐45‐1970**Tastiness**
Positive(D)LPFC (peak in SFG)L/R5.732795183546Visual CortexR5.2199245‐8213Negative‐L4.49415‐54‐7322

Results are reported at a statistical threshold of p<0.05, FWE corrected at cluster-‐level (height threshold of p<0.001), only peak activations of clusters are reported; L=left hemisphere, R=right hemisphere, MNI=Montreal Neurological Institute, k=cluster size in voxels.

Importantly, for both choice-relevant attributes in the food task, we found that the cluster in the DLPFC closely matched the cluster identified in our previous approach (see Author response image 8 for illustration of overlap). The close match of identified brain regions for goal-dependent coding of attribute values provides further evidence for robustness of our previous evidence.

**Author response image 8. respfig8:** 

On a more general level, these findings suggest that decoding accuracies in goal-dependent brain regions reflect individual differences in attribute weights (across subjects and conditions), rather than only overall changes in signal‐to-noise due to attention to an attribute.

Next, we implemented a similar analysis for the altruism task. Consistent with our results suggesting that Altruistic Choice may be a more heterogeneous process, results of the regression analysis for $Self, $Other, and Fairness were weaker and did not survive correction for multiple comparisons in this whole- brain analyses. However, at a slightly more lenient statistical threshold (p < 0.001, uncorrected), neural information in the right DLPFC on trial-wise values of $Self indeed reflected individual weights assigned to $Self during choices as estimated in the behavioral model (w $Self). This result further supports the notion that neural activation patterns in the right DLPFC are involved in goal-consistent coding for peoples own monetary benefits in the altruism task. A matching analyses that tested for brain regions in which neural information is negatively correlated with attribute weights for $Self did not yield any significant results at this statistical threshold. See Author Response Table 2 for complete list of results for $Self.

Taken together with the results of the regression analyses for food attributes reported above, these supplemental analyses provide further support for the notion that flexible attribute representations in the right DLPFC generalize across task domains (i.e., $Self, Tastiness, Healthiness). Please see Author response image 9 for an illustration of the overlap of the DLPFC cluster identified in the whole brain regression analyses (green, for illustration purposes displayed at p < 0.005) with the DLPFC cluster identified in the conjunction analysis of goal-consistent attribute representations of regulatory vs. natural task conditions (orange).

**Author response image 9. respfig9:** 

Similar whole-brain regression analyses were implemented for Fairness and $Other (for complete list of results at p < 0.001, uncorrected, see Author response table 2). Interestingly, for $Other, we found that at this threshold neural information on $Other significantly covaried with the subject’s w$Other in the left pSTS/TPJ (while the right pSTS/TPJ only became significant at a threshold of p < 0.005, uncorrected). For illustration of both clusters (and the PCC/Precunes) at p < 0.005, uncorrected, see panel A). Notably, the cluster in the right pSTS/TPJ closely matched the brain region identified using our previous approach in which we compared regulatory and natural conditions (see panel B for illustration). While results at this liberal statistical threshold have to be interpreted with extreme caution, these findings suggest that goal-sensitive neural representations of $Other might be encoded bilaterally in the pSTS/TPJ.

Taken together, these results suggest that our results are largely robust to different ways of conceptualizing the correspondence between decoding accuracies and behaviorally estimated attribute weights. As mentioned above, we have chosen to report the original ANOVAs in the paper because of their greater flexibility in identifying patterns that do not fully conform to the predictions of the behavioral model. We hope that the reviewer’s concerns on this regard have been fully addressed.

**Author response image 10. respfig10:** 

Author response table 2: Brain regions in which neural decoding accuracies for trial-wise attribute values covaried with the weights (estimated in the DDM) in the altruism task across conditions.

AttributeBrain regionSideTkMNI xyz***$Self***Positive(D)LPFCR3.889361134Superior Frontal Gyrus/L3.568-92631Paracingulate CortexVisual CortexR3.691112-974PrecuneusR3.832218-4643CerebellumL4.4867-12-70-47CerebellumL4.4867-12-70-47Negative-***Fairness***PositiveInferior Parietal Cortex *R4.0319257-5852Inferior Frontal Gyrus/Rolandic operculumR3.341257516ThalamusL3.6615-9-410Inferior Temporal GyrusL3.6929-51-31-20Visual CortexL/R3.83130-3-701CerebellumL/R3.61220-46-17Angular GyrusR3.511627-6746Negative-***$Other***PositivePosterior Superior Temporal GyrusL854.30-42-6110DMPFCL483.95-63870R63.18184743Inferior Temporal GyrusR223.6563-40-29CerebellumL/R1113.843-46-8Visual CortexL/R93.363-7322Negative-

Results are reported at a statistical threshold of p<0.001 (uncorrected) with a cluster-size threshold of k > 5 voxels, only peak activations of clusters within brain mask are reported; * indicates FWE corrected at cluster-level; L=left hemisphere, R=right hemisphere, MNI=Montreal Neurological Institute, k=cluster size in voxels.